# Angle Domain Guidance: Latent Diffusion Requires Rotation Rather Than Extrapolation

**Cheng Jin** [1]  **Zhenyu Xiao** [*1]  **Chutao Liu** [*2]  **Yuantao Gu** [1]

## Abstract

Classifier-free guidance (CFG) has emerged as a pivotal advancement in text-to-image latent diffusion models, establishing itself as a cornerstone technique for achieving high-quality image synthesis. However, under high guidance weights, where text-image alignment is significantly enhanced, CFG also leads to pronounced color distortions in the generated images. We identify that these distortions stem from the amplification of sample norms in the latent space. We present a theoretical framework that elucidates the mechanisms of norm amplification and anomalous diffusion phenomena induced by classifier-free guidance. Leveraging our theoretical insights and the latent space structure, we propose an Angle Domain Guidance (ADG) algorithm. ADG constrains magnitude variations while optimizing angular alignment, thereby mitigating color distortions while preserving the enhanced text-image alignment achieved at higher guidance weights. Experimental results demonstrate that ADG significantly outperforms existing methods, generating images that not only maintain superior text alignment but also exhibit improved color fidelity and better alignment with human perceptual preferences.

## 1. Introduction

The Stable Diffusion series (Rombach et al., 2022; Esser et al., 2024), along with other latent-diffusion-based generative models (Ramesh et al., 2022; Labs, 2024), has revolutionized text-to-image generation and related downstream tasks (Zhang et al., 2023; Zhao et al., 2024; Zhang et al.,

2024). These models operate by performing diffusion in the latent space, which is then decoded into the image space, effectively transforming textual descriptions into high-quality images with intricate details and compelling visual effects. A key component of this process is the Classifier-Free Guidance (CFG) mechanism (Ho & Salimans, 2021), which enhances generation quality through linear extrapolation of the conditional score function with a guidance weight.

Despite its effectiveness, CFG introduces several challenges. Research has shown that increasing the guidance weight improves text alignment but at the cost of reduced sample diversity (Ho & Salimans, 2021). More critically, when the guidance weight surpasses a certain threshold, the quality of the generated images deteriorates significantly, leading to a loss of detail and overall visual degradation. Theoretically, this phenomenon may be attributed to the non-commutativity between the tilting process, driven by the conditional likelihood, and the noise addition process (Wu et al., 2024; Chidambaram et al., 2024). This non-commutativity results in a deviation of the CFG-generated samples from the target distribution, manifesting as performance degradation, particularly at high guidance weights.

Several approaches have been proposed to mitigate these issues. Techniques such as variable weighting schemes and additional Langevin sampling iterations have been introduced to counteract the decline in image quality (Chung et al., 2025; Xia et al., 2024; Bradley & Nakkiran, 2024; Sadat et al., 2025). While these methods have shown some efficacy in reducing quality degradation for specific tasks, they remain constrained by their reliance on a linear combination of conditional score functions. This linear framework limits their ability to maintain generation quality, highlighting the need for a more comprehensive solution.

In this paper, we first observe that higher guidance weights are associated with larger sample norms in latent space, leading to high image saturation and color distortion. We then provide a theoretical analysis of the CFG mechanism, demonstrating that the linear extrapolation of the score function inevitably causes norm amplification and anomalous diffusion behavior. Building on this analysis, we propose an Angle Domain Guidance (ADG) algorithm, inspired by the high-dimensional Gaussian assumption in latent space

---

[*]Equal contribution  [1]Department of Electronic Engineering, Tsinghua University, Beijing, China [2]Zhili College, Tsinghua University, Beijing, China. Correspondence to: Yuantao Gu <gyt@tsinghua.edu.cn>.

*Proceedings of the 42$^{nd}$ International Conference on Machine Learning*, Vancouver, Canada. PMLR 267, 2025. Copyright 2025 by the author(s).

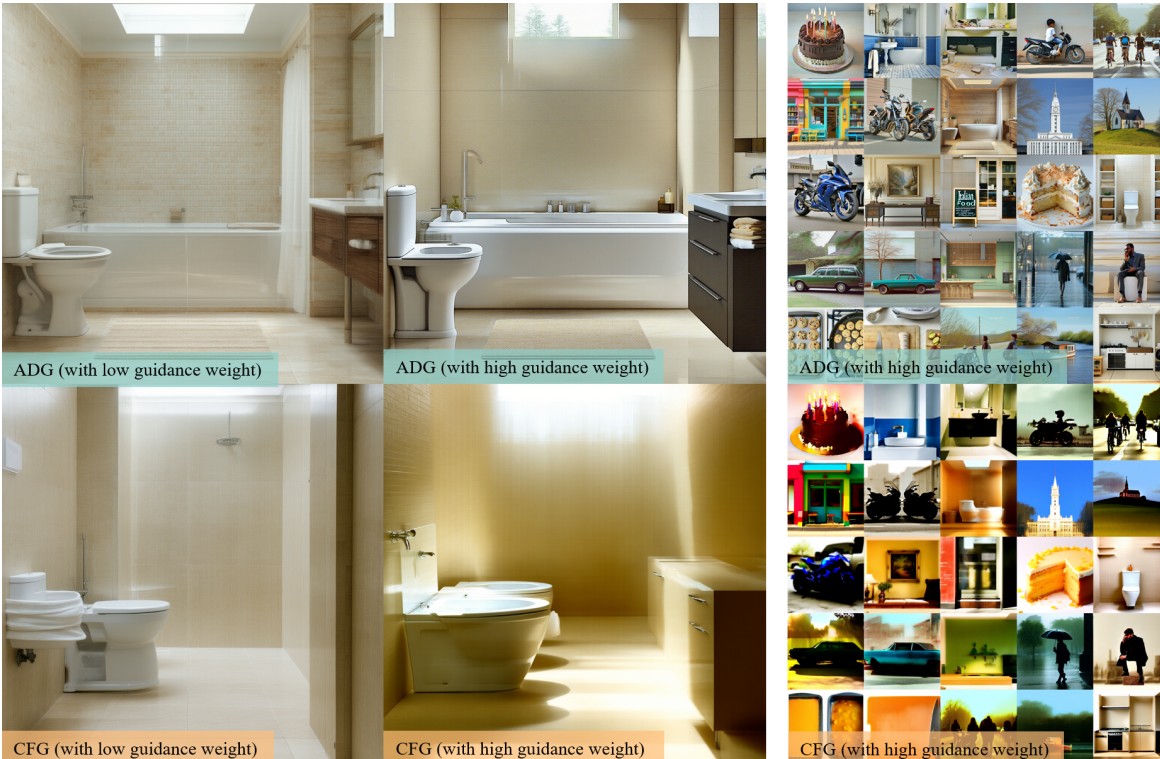

*Figure 1.* Comparison of ADG and CFG using the same random seed with the prompt "***A bathroom with a toilet and a bathtub.***" At low guidance weights, ADG accurately aligns with the prompt, while CFG fails to do so. At high guidance weights, ADG remains stable, whereas CFG suffers from abnormal saturation and color distortion.

inherent in variational autoencoders. The ADG algorithm transforms the score function into an expectation over a time-dependent distribution, which is then enhanced in the angular domain before being re-mapped back into the score function for diffusion. Experimental results on the COCO dataset demonstrate that the ADG algorithm outperforms the baseline, particularly in generating images that are better aligned with textual descriptions, underscoring its potential to overcome the limitations of existing methods. The implementation is available at github.com/jinc7461/ADG.

## 2. Background

### 2.1. Conditional Diffusion Model

Diffusion models are composed of two fundamental processes: a forward process that gradually transforms the target distribution into Gaussian noise, and a reverse process that reconstructs the target distribution from the noise distribution (Song et al., 2021b). The forward process is governed by the following stochastic differential equation (SDE):

$$\mathrm{d}\boldsymbol{x}_t = \boldsymbol{f}(\boldsymbol{x}_t, t)\mathrm{d}t + g(t)\mathrm{d}\boldsymbol{w}, \quad \boldsymbol{x}_0 \sim p_0(\cdot|\boldsymbol{c}), \quad (1)$$

where $\boldsymbol{w}$ denotes the Wiener process, and $p_0(\cdot|\boldsymbol{c})$ represents the conditional target distribution, with $\boldsymbol{c}$ denoting a condition (such as text in text-to-image models). Specifically, $p_0(\cdot|\emptyset)$ represents the unconditional distribution.

In this work, we adopt the widely used Variance Preserving (VP) SDE (Song et al., 2021b) for the forward process:

$$\mathrm{d}\boldsymbol{x}_t = -\frac{\beta(t)}{2}\boldsymbol{x}_t\mathrm{d}t + \sqrt{\beta(t)}\mathrm{d}\boldsymbol{w}, \quad \boldsymbol{x}_0 \sim p_0(\cdot|\boldsymbol{c}), \quad (2)$$

where the dynamics ensure that $p_T$ approximates a standard Gaussian distribution for sufficiently large $T$. The transition distribution of $\boldsymbol{x}_t$ conditioned on $\boldsymbol{x}_0$ is given by:

$$\boldsymbol{x}_t|\boldsymbol{x}_0 \sim \mathcal{N}(\sqrt{\bar{\alpha}_t}\boldsymbol{x}_0, \bar{\beta}_t\mathbf{I}), \quad (3)$$

where $\bar{\alpha}_t = \exp\left(\int_0^t -\beta(\tau)\,\mathrm{d}\tau\right)$ and $\bar{\beta}_t = 1 - \bar{\alpha}_t$.

The reverse process, which reconstructs the target distribution, corresponds to the time-reversed SDE:

$$\mathrm{d}\tilde{\boldsymbol{x}}_t = \left[-\frac{\beta(t)}{2}\tilde{\boldsymbol{x}}_t - \beta(t)\nabla_{\tilde{\boldsymbol{x}}_t}\log p_t(\tilde{\boldsymbol{x}}_t|\boldsymbol{c})\right]\mathrm{d}t + \sqrt{\beta(t)}\mathrm{d}\tilde{\boldsymbol{w}}, \quad (4)$$

where $\tilde{\boldsymbol{w}}$ represents the reverse Wiener process, $p_t$ is the distribution of $\boldsymbol{x}_t$ in (2), and $\nabla\log p_t$ is referred to as the

*score function*. Alternatively, the sampling process can be described by the following reverse-time ordinary differential equation (ODE):

$$\mathrm{d}\bar{\boldsymbol{x}}_t = \left[ -\frac{\beta(t)}{2}\bar{\boldsymbol{x}}_t - \frac{\beta(t)}{2}\nabla_{\bar{\boldsymbol{x}}_t}\log p_t(\bar{\boldsymbol{x}}_t|\boldsymbol{c}) \right]\mathrm{d}t, \quad (5)$$

which eliminates the stochastic noise sampling during the process, making it more commonly used in practice.

According to classical results from probability theory (Anderson, 1982), if the reverse-time process—either SDE or ODE—is initialized with $q_T = p_T$, then the marginal distributions match at all times, i.e., $q_t = p_t$ for all $t \in [0, T]$. Consequently, if it is possible to sample from $p_T$ and obtain the score function $\nabla_{\boldsymbol{x}}\log p_t(\boldsymbol{x}|\boldsymbol{c})$ for different times $t$, one can generate samples from the target distribution $p_0$ by discretizing either (4) or (5).

Since the score function $\nabla_{\boldsymbol{x}}\log p_t(\boldsymbol{x}|\boldsymbol{c})$ is generally intractable and $p_T$ is approximated as a standard Gaussian distribution, practical first-order discretization methods for (4) and (5) correspond to the DDPM sampler (Ho et al., 2020) and DDIM sampler (Song et al., 2021a), respectively:

$$\boldsymbol{x}_{t-\Delta t} = \sqrt{\frac{\bar{\alpha}_{t-\Delta t}}{\bar{\alpha}_t}}\boldsymbol{x}_t + \left( 1 - \frac{\bar{\alpha}_t}{\bar{\alpha}_{t-\Delta t}} \right)\boldsymbol{s}_\theta(\boldsymbol{x}_t, t, \boldsymbol{c})$$
$$+ \sqrt{1 - \frac{\bar{\alpha}_t}{\bar{\alpha}_{t-\Delta t}}}\boldsymbol{\eta}, \quad (6)$$

$$\boldsymbol{x}_{t-\Delta t} = \sqrt{\frac{\bar{\alpha}_{t-\Delta t}}{\bar{\alpha}_t}}\boldsymbol{x}_t + \frac{1}{2}\left( 1 - \frac{\bar{\alpha}_t}{\bar{\alpha}_{t-\Delta t}} \right)\boldsymbol{s}_\theta(\boldsymbol{x}_t, t, \boldsymbol{c}), \quad (7)$$

where $\boldsymbol{s}_\theta(\boldsymbol{x}_t, t, \boldsymbol{c})$ is a neural network trained to approximate the score function, $\boldsymbol{\eta} \sim \mathcal{N}(\boldsymbol{0}, \mathbf{I})$, and $\boldsymbol{x}_T \sim \mathcal{N}(\boldsymbol{0}, \mathbf{I})$.

To enhance the stability of neural network training, in practice, the network is trained to approximate $-\sqrt{\bar{\beta}_t}\nabla\log p_t$ by $\boldsymbol{\epsilon}_\theta$, rather than directly approximating $\nabla\log p_t$ by $\boldsymbol{s}_\theta$. This approach mitigates numerical instabilities and improves convergence during training.

## 2.2. Classifier-Free Guidance

While the theoretical foundations of vanilla conditional sampling based on diffusion models are well-established, directly using (4) or (5) for generation often results in low-quality outputs (Ho & Salimans, 2021). To address this limitation, the Classifier-Free Guidance (CFG) technique was introduced. CFG aims to sample from a modified distribution $p_{0,\omega}(\boldsymbol{x}) \propto p_0(\boldsymbol{x})p^\omega(\boldsymbol{c}|\boldsymbol{x})$, which represents the unconditional data distribution tilted by the conditional likelihood. Using Bayes' theorem, this distribution can be reformulated as:

$$p_{0,\omega}(\boldsymbol{x}) \propto p_0^{1-\omega}(\boldsymbol{x})p_0^\omega(\boldsymbol{x}|\boldsymbol{c}),$$

and the gradient of the log-probability becomes:

$$\nabla\log p_{0,\omega}(\boldsymbol{x}) = (1-\omega)\nabla\log p_0(\boldsymbol{x}) + \omega\nabla\log p_0(\boldsymbol{x}|\boldsymbol{c}).$$

Inspired by this formulation, the Classifier-Free Guidance (CFG) method replaces the score function $\nabla_{\bar{\boldsymbol{x}}_t}\log p_t(\bar{\boldsymbol{x}}_t|\boldsymbol{c})$ with a weighted combination of the unconditional and conditional score functions, denoted as $\boldsymbol{s}_{\mathrm{cfg},\omega}(\bar{\boldsymbol{x}}_t, t, \boldsymbol{c})$, defined as:

$$\boldsymbol{s}_{\mathrm{cfg},\omega}(\bar{\boldsymbol{x}}_t, t, \boldsymbol{c}) = (1-\omega)\nabla_{\bar{\boldsymbol{x}}_t}\log p_t(\bar{\boldsymbol{x}}_t|\emptyset)$$
$$+ \omega\nabla_{\bar{\boldsymbol{x}}_t}\log p_t(\bar{\boldsymbol{x}}_t|\boldsymbol{c}), \quad (8)$$

in the reverse process. For the ODE sampler, this process can be expressed as:

$$\mathrm{d}\bar{\boldsymbol{x}}_t = [-\bar{\boldsymbol{x}}_t - \boldsymbol{s}_{\mathrm{cfg},\omega}(\bar{\boldsymbol{x}}_t, t, \boldsymbol{c})]\,\mathrm{d}t. \quad (9)$$

In the above equation, $\omega$, referred to as the guidance weight, quantifies the strength of the tilting toward the conditional distribution which is always bigger than 1. However, due to the non-commutativity of the tilting process with the forward process, for any $t > 0$,

$$\nabla\log p_{t,\omega}(\boldsymbol{x}) \neq \boldsymbol{s}_{\mathrm{cfg},\omega}(\boldsymbol{x}, t, \boldsymbol{c}), \quad (10)$$

where $p_{t,\omega}(\boldsymbol{x})$ is the distribution of $\boldsymbol{x}_t$ in (2) with $p_0(\cdot|\boldsymbol{c})$ replaced by $p_{t,\omega}$. This discrepancy implies that CFG does not strictly correspond to a sampling process targeting $p_{0,\omega}$ as the true target distribution.

## 3. Revisiting Classifier-Free Guidance

### 3.1. Norm Amplification in CFG

While Classifier-Free Guidance (CFG) effectively addresses the limitations of vanilla conditional diffusion models, it introduces notable side effects. Specifically, increasing the guidance weight has a significant impact on the generated samples: they exhibit larger $\ell_2$-norms, and the corresponding decoded images show higher saturation levels, often accompanied by distortion. Figure 2 illustrates this phenomenon, depicting how the $\ell_2$-norm of $\boldsymbol{x}_0$ varies with changes in the guidance weight, and the relationship between the latent variable norm and image saturation.

This phenomenon can be understood through the interplay between the conditional and unconditional distributions. When CFG operates on a conditional probability distribution that aligns with the "surface" of the unconditional distribution, the generated samples tend to deviate further from the overall distribution, resulting in larger norms. To formalize this, we consider the case of a Gaussian mixture model:

$$p_0(\boldsymbol{x}) = \sum_{c=1}^{C} \pi_c \mathcal{N}(\boldsymbol{x}|\boldsymbol{\mu}_c, \mathbf{I}), \quad (11)$$

where $\pi_c$ denotes the mixing coefficient satisfying $\sum_{c=1}^{C} \pi_c = 1$, and the conditional distribution is given by:

$$p_0(\boldsymbol{x}|c) = \mathcal{N}(\boldsymbol{x}|\boldsymbol{\mu}_c, \mathbf{I}).$$

**Definition 3.1** (surface class). *A class $c^*$ is a surface class if there exists a hyperplane defined by $\boldsymbol{w}^\top \boldsymbol{x} + b = 0$ such that $\boldsymbol{w}^\top \boldsymbol{\mu}_{c^*} + b = 0$, and for all $c \neq c^*$, $(\boldsymbol{w}^\top \boldsymbol{\mu}_c + b) < 0$, indicating that $\boldsymbol{\mu}_c$ lies strictly on the same side of the hyperplane and $\boldsymbol{w}$ points to the outer side of the uncoditional distribution.*

We focus on a surface class $c^*$, defined as a class whose mean $\boldsymbol{\mu}_{c^*}$ lies at a vertex of the polytope formed by $\{\boldsymbol{\mu}_c\}_{c=1}^{C}$. This concept is formalized in Definition 3.1.

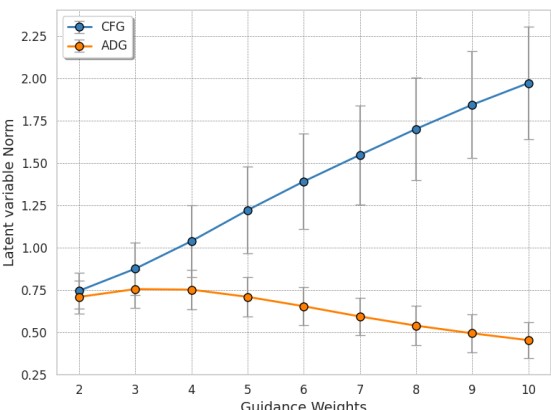

2.a Latent Variable Norm with Different Guidance Weights

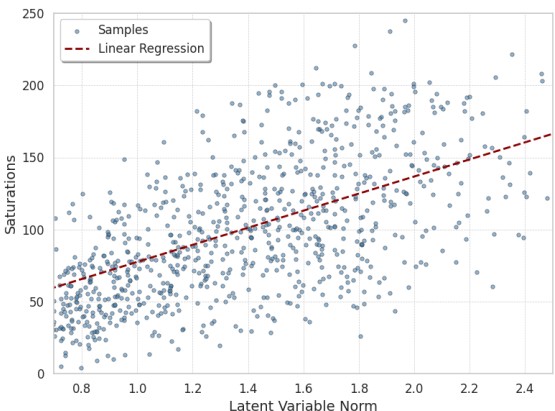

2.b Relationship Between Latent Variable Norm and Image Saturation

*Figure 2.* Average latent variable norm ($\ell_2$) and image saturation under different guidance weights using CFG with SD3.5 (d=38). Prompts are randomly selected from 100 samples in the COCO dataset. As the guidance weight increases, the norm of the hidden space samples generated by the CFG algorithm increases, which leads to oversaturation of the corresponding images.

Samples generated by CFG diffusion (9) are more likely to

move toward the outer regions of the distribution compared to samples generated by standard conditional diffusion (5). This tendency is captured in Theorem 3.2, which quantifies the norm amplification effect.

**Theorem 3.2** (Norm amplification). *For the same initial point $\boldsymbol{x}_T$, let $\hat{\boldsymbol{x}}_0$ denote the sample obtained using the CFG sampler (9), and $\boldsymbol{x}_0$ denote the sample obtained using the ODE sampler (5). Then:*

$$\boldsymbol{w}^\top \hat{\boldsymbol{x}}_0 > \boldsymbol{w}^\top \boldsymbol{x}_0, \tag{12}$$

*indicating that CFG-sampled points align more strongly with the outer direction defined by $\boldsymbol{w}$.*

The result highlights how CFG encourages samples to move toward outer regions, with their $\ell_2$-norms growing as the guidance weight increases. Figure 3 visualizes this behavior, showing how samples deviate from the data distribution under different guidance weights.

### 3.2. Anomalous Diffusion in CFG

Building on Theorem 3.2, we further investigate the geometric and probabilistic properties of CFG sampling in the latent space. Specifically, we define a family of time-dependent latent manifolds $\mathcal{M}_t$ as:

$$\mathcal{M}_t = \{\boldsymbol{x} \mid \boldsymbol{s}_{\text{cfg},\omega}^\top(\boldsymbol{x}, t, c^*) \nabla \log p_t(\boldsymbol{x}|c^*) \leq 0\}. \tag{13}$$

Regions defined by $\mathcal{M}_t$ exhibit anomalous diffusion, where the update direction of sampled points tends toward areas with lower probability density.

**Theorem 3.3** (Anomalous diffusion). *For the unit normal vector $\boldsymbol{w}$ of the hyperplane in Definition 3.1, there exists a constant $C_1 > 0$, dependent on $p_0$, $\omega$, and $t$, such that:*

$$\bar{\alpha}_t \boldsymbol{\mu}_{c^*} + k\boldsymbol{w} \in \mathcal{M}_t, \quad \forall k \in (0, C_1]. \tag{14}$$

*Furthermore, the constant $C_1$ increases as the guidance weight $\omega$ become larger.*

Theorem 3.3 establishes that CFG sampling induces anomalous diffusion phenomena within the neighborhood extending beyond the boundaries of the conditional distribution. As the guidance weight $\omega$ increases, the anomalous regions defined by $\mathcal{M}_t$ expand, amplifying the deviation of CFG-generated samples from the original conditional distribution.

To provide theoretical insight while maintaining conciseness, we outline the proof of Theorem 3.3. First, we characterize the score function via Lemma 3.4.

**Lemma 3.4** (Lemma 1 of (Huang et al., 2023)). *For any $t \in (0, T]$, the score function is given by:*

$$\nabla_{\boldsymbol{x}} \log p_t(\boldsymbol{x}|\boldsymbol{c}) = \frac{\sqrt{\bar{\alpha}_t}\hat{\boldsymbol{x}}_0^{(\boldsymbol{c})} - \boldsymbol{x}}{\bar{\beta}_t}, \tag{15}$$

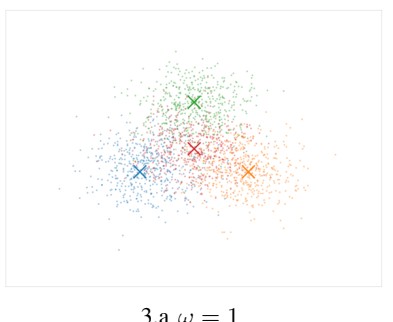
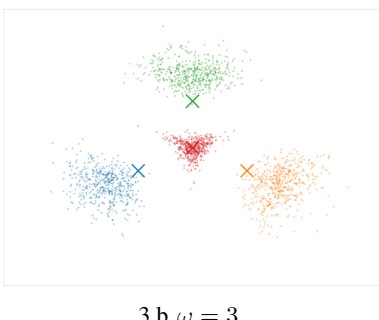
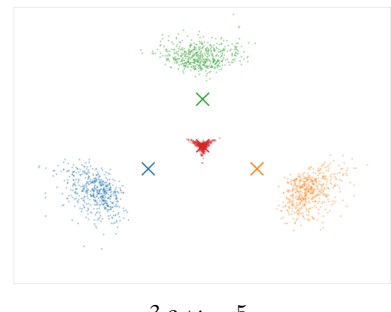

3.a $\omega = 1$     3.b $\omega = 3$     3.c $\omega = 5$

*Figure 3.* Generated sample distributions of a four-component Gaussian mixture model under varying guidance weights. True component means are marked, and samples corresponding to surface classes move further from the distribution as $\omega$ increases.

*where*

$$\hat{\boldsymbol{x}}_0^{(\boldsymbol{c})} = \mathbb{E}_{\boldsymbol{x}_0 \sim q_{\boldsymbol{c},t,\boldsymbol{x}}}[\boldsymbol{x}_0],$$

$$q_{\boldsymbol{c},t,\boldsymbol{x}}(\boldsymbol{x}_0) \propto p_0(\boldsymbol{x}_0|\boldsymbol{c}) \exp\left(-\frac{\|\boldsymbol{x} - \sqrt{\bar{\alpha}_t}\boldsymbol{x}_0\|_2^2}{2\bar{\beta}_t}\right).$$

Lemma 3.4 shows that the score function can be characterized by the expectation of the conditional distribution tilted by a time-dependent Gaussian distribution. Using this result, $\boldsymbol{s}_{\text{cfg},\omega}$ can be rewritten as:

$$\boldsymbol{s}_{\text{cfg},\omega} = (\omega - 1)\frac{\bar{\alpha}_t(\hat{\boldsymbol{x}}_0^{(\boldsymbol{c})} - \hat{\boldsymbol{x}}_0^{(\emptyset)})}{\bar{\beta}_t}$$
$$+ \frac{\bar{\alpha}_t\hat{\boldsymbol{x}}_0^{(\emptyset)} - \boldsymbol{x}}{\bar{\beta}_t}. \tag{16}$$

The distinction between $\boldsymbol{s}_{\text{cfg},\omega}$ and $\nabla_{\boldsymbol{x}} \log p_t(\boldsymbol{x}|c^*)$ lies in the first term of (16). For a point $\boldsymbol{x} = \bar{\alpha}_t\boldsymbol{\mu}_{c^*} + k\boldsymbol{w}$ with sufficiently small $k$, we find:

$$\boldsymbol{w}^\top(\hat{\boldsymbol{x}}_0^{(\boldsymbol{c})} - \hat{\boldsymbol{x}}_0^{(\emptyset)}) > \epsilon, \tag{17}$$

where $\epsilon > 0$. Further, noting that $\nabla_{\boldsymbol{x}} \log p_t(\boldsymbol{x}|c^*) = -k\boldsymbol{w}$, we have:

$$\boldsymbol{s}_{\text{cfg},\omega}^\top(\boldsymbol{x},t,c^*)\nabla \log p_t(\boldsymbol{x}|c^*) < -\epsilon k + k^2, \tag{18}$$

thereby proving the original proposition.

From the analysis, it can be seen that the CFG method is equivalent to enhancing $\hat{\boldsymbol{x}}_0^{(\boldsymbol{c})}$ to $\hat{\boldsymbol{x}}_0^{(\boldsymbol{c})} + (\omega - 1)(\hat{\boldsymbol{x}}_0^{(\boldsymbol{c})} - \hat{\boldsymbol{x}}_0^{(\emptyset)})$. While this enhancement effectively amplifies features that align with the target distribution, it unintentionally leads to excessively large norms, causing oversaturation and distortion artifacts when the guidance weight is high.

## 4. Angle-Domain Guidance Sampling (ADG)

Building upon the insights presented in Section 3, we identify Classifier-Free Guidance (CFG) as an operation that reinforces $\hat{\boldsymbol{x}}_0$ linearly, leading to generated samples with excessively high norms. While CFG effectively captures features aligned with the target distribution, it also inflates sample magnitudes, resulting in oversaturation, artifacts, and unrealistic images at high guidance weights. To mitigate these issues, we propose the Angle-Domain Guidance Sampling (ADG) method, which focuses on the directional alignment of samples with the target while limiting changes in their magnitudes. A simplified version of ADG is provided in Appendix E as Algorithm 8.

### 4.1. Motivation for ADG

The primary motivation underlying ADG is that the directional information encoded in $\hat{\boldsymbol{x}}_0$ is often sufficient to guide samples toward the target distribution. Latent space diffusion models are typically based on the assumption of a high-dimensional isotropic Gaussian inherent in variational autoencoders (Kingma, 2013), which concentrates around a spherical shell with a fixed radius (Wainwright, 2019). Although due to limitations in network capacity, training data, and other factors, the latent space distribution cannot be fully modeled by a high-dimensional Gaussian, this assumption still offers valuable insights. Specifically, CFG can be seen as enhancing $\hat{\boldsymbol{x}}_0$ in the linear domain, while emphasizing the differences in both the angle and magnitude between $\hat{\boldsymbol{x}}_0^{(\boldsymbol{c})}$ and $\hat{\boldsymbol{x}}_0^{(\emptyset)}$. While norm adjustments may marginally improve semantic alignment, excessive norm amplification at high guidance weights becomes detrimental, resulting in oversaturation and distortion. Therefore, ADG shifts the focus to differences in the angular domain, while constraining variations in the magnitude domain.

### 4.2. Methodology

The proposed ADG algorithm is comprehensively described in Algorithm 1. Figure 5 illustrates the distinct behaviors of $\hat{\boldsymbol{x}}_0$ under ADG compared to CFG, emphasizing the enhanced stability and directional control achieved by ADG.

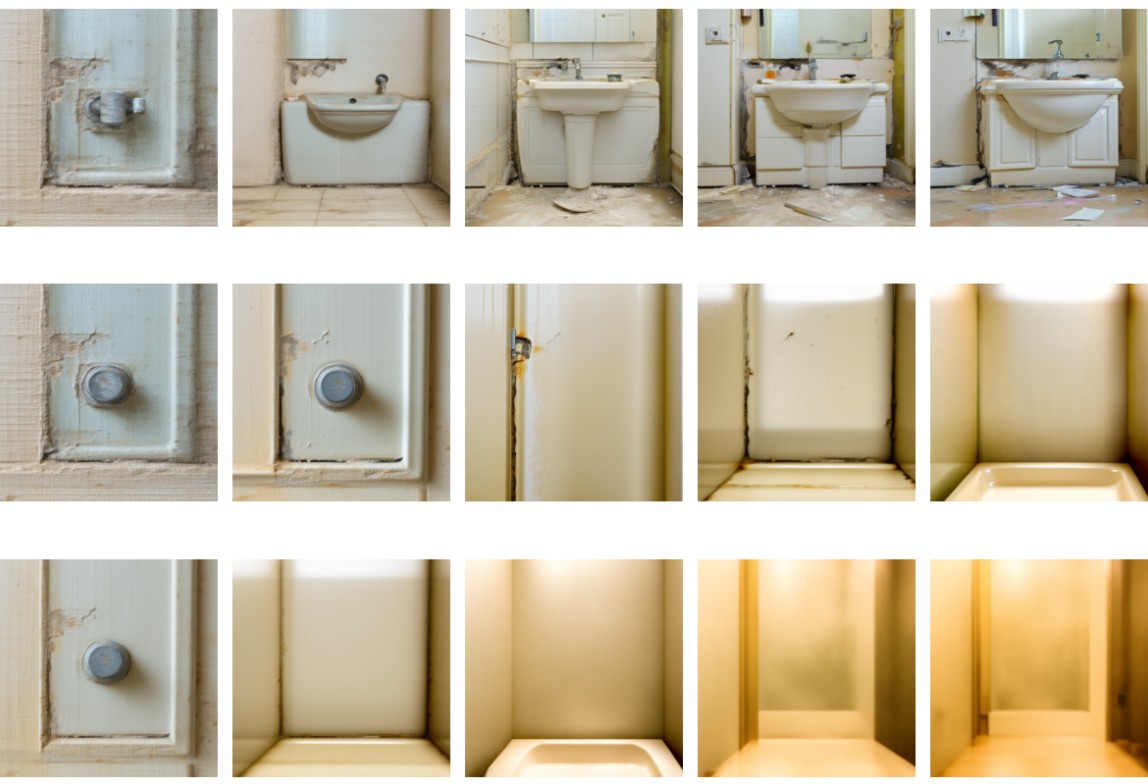

*Figure 4.* Comparison of generated images using different guidance strategies for the prompt "***Part of a small bathroom in need of repair.***" The rows correspond to different guidance algorithms: ADG (top), CFG (middle), and CFG++ (bottom). The columns represent increasing guidance weights: 2, 4, 6, 8, and 10 (from left to right).

Unlike CFG, which often leads to exaggerated magnitudes in $\hat{x}_0$, ADG introduces a refined mechanism that prioritizes alignment with direction. Moreover, Proposition 4.1 formalizes the constraints imposed by ADG on the magnitude of $\hat{x}_0$. These constraints ensure a balanced trade-off between directional guidance and structural integrity, resulting in improved sample quality.

---

**Algorithm 1** Angle-Domain Guidance Sampling (ADG)

**Require:** $x_T \sim \mathcal{N}(0, \mathbf{I}), 1 < \omega \in \mathbb{R}$, Decoder $\mathcal{D}$

**for** $t = T$ **to** $1$ **do**

$\quad \hat{x}_0^{(c)} = (x_t - \sqrt{1 - \bar{\alpha}_t}\epsilon_\theta(x, t, c))/\sqrt{\bar{\alpha}_t}$

$\quad \hat{x}_0^{(\emptyset)} = (x_t - \sqrt{1 - \bar{\alpha}_t}\epsilon_\theta(x, t, \emptyset))/\sqrt{\bar{\alpha}_t}$

$\quad \gamma = \arccos\left(\frac{(\hat{x}_0^{(\emptyset)})^\top \hat{x}_0^{(c)}}{\|\hat{x}_0^{(\emptyset)}\|_2\|\hat{x}_0^{(c)}\|_2}\right)$

$\quad \gamma_\omega = \text{threshold}((\omega - 1)\gamma, \pi/3)$

$\quad \hat{x}_{0,\omega} = \cos(\gamma_\omega)\hat{x}_0^{(c)} + \frac{\sin(\gamma_\omega)}{\sin(\gamma)}(\hat{x}_0^{(c)} - \text{proj}_{\hat{x}_0^{(\emptyset)}}(\hat{x}_0^{(c)}))$

$\quad x_{t-1} = \sqrt{\bar{\alpha}_{t-1}}\hat{x}_{0,\omega} + \sqrt{1 - \bar{\alpha}_{t-1}}\frac{x_t - \bar{\alpha}_t \hat{x}_{0,\omega}}{\sqrt{1 - \bar{\alpha}_t}}$

**end for**

$I = \mathcal{D}(x_0)$

---

**Proposition 4.1.** *For any* $t \in (0, T]$, $\hat{x}_{0,\omega}$ *and* $\hat{x}_0^{(c)}$ *defined in Algorithm 1 satisfy the following property:*

$$\|\hat{x}_{0,\omega}\|_2 \leq \sqrt{2}\|\hat{x}_0^{(c)}\|_2. \qquad (19)$$

ADG demonstrates remarkable flexibility, making it adaptable to a wide range of sampling frameworks. It is compatible with advanced deterministic samplers, such as DPM-Solver (Lu et al., 2022a;b), as well as stochastic samplers like DDPM. This versatility arises from the fact that all sampling algorithms can be expressed as weighted combinations of $x_t$, $\hat{x}_0$, and Gaussian noise. By replacing the original $\hat{x}_0$ with $\hat{x}_{0,\omega}$ derived from ADG, the method seamlessly generalizes across various diffusion-based approaches. Additionally, ADG extends naturally to flow-matching generative models (Lipman et al., 2023), which are mathematically equivalent to diffusion models with some assumptions(Patel et al., 2024). Detailed derivations of ADG's extension to flow-matching models are provided in Appendix F.

## 5. Experiment

### 5.1. Text-To-Image Task

To evaluate the effectiveness of our proposed method, we conduct experiments using Stable Diffusion v3.5 (d=38)

*Table 1.* Results of 10 NFE generation with SD v3.5 (d=38) on COCO10k, best performance includes finer grained data.

| METHOD | $\omega = 2$ | | | $\omega = 4$ | | | $\omega = 6$ | | |
|---|---|---|---|---|---|---|---|---|---|
| | CLIP ↑ | IR ↑ | FID ↓ | CLIP ↑ | IR ↑ | FID ↓ | CLIP ↑ | IR ↑ | FID ↓ |
| ADG (PROPOSED) | 0.315 | **0.727** | 17.4 | **0.319** | **0.928** | 17.1 | **0.321** | **0.964** | **16.9** |
| CFG | **0.316** | 0.711 | 17.5 | 0.318 | 0.835 | **16.9** | 0.317 | 0.586 | 17.1 |
| CFG++ ($\lambda = \omega/12.5$) | 0.315 | 0.574 | **17.1** | 0.306 | -0.148 | 18.0 | 0.282 | -0.980 | 19.8 |
| APG | 0.315 | 0.651 | 17.5 | 0.317 | 0.910 | 17.2 | 0.319 | **0.964** | 16.9 |
| | $\omega = 10$ | | | $\omega = 15$ | | | BEST PERFORMANCE | | |
| | CLIP ↑ | IR ↑ | FID ↓ | CLIP ↑ | IR ↑ | FID ↓ | CLIP ↑ | IR ↑ | FID ↓ |
| ADG (PROPOSED) | **0.322** | **0.970** | 16.6 | **0.324** | **0.940** | 16.8 | **0.324** | **0.970** | 16.6 |
| CFG | 0.304 | -0.142 | 17.7 | 0.290 | -0.659 | 20.3 | 0.318 | 0.843 | 16.6 |
| CFG++($\lambda = \omega/12.5$) | 0.257 | -1.509 | 21.4 | 0.246 | -1.680 | 22.6 | 0.315 | 0.574 | 17.1 |
| APG | 0.318 | 0.935 | **16.4** | 0.316 | 0.691 | **16.1** | 0.319 | 0.966 | **16.1** |

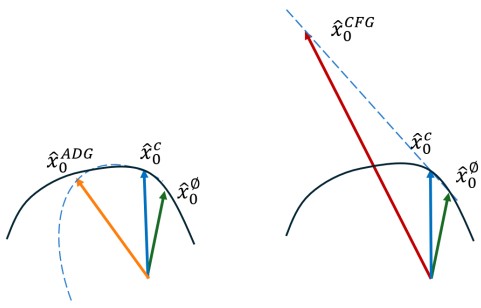

*Figure 5.* Comparison of angular domain guidance (ADG) and classifier-free guidance (CFG). The left diagram illustrates the ADG update, which preserves the latent variable norm by focusing on angular adjustments, while the right diagram shows the CFG, which amplifies the norm due to direct linear adjustments. The black line represents the potential manifold structure.

on the COCO dataset(Lin et al., 2014), comparing against CFG (Ho & Salimans, 2021), CFG++ (Chung et al., 2025), and APG (Sadat et al., 2025). For CFG++ and APG, we adopt the recommended parameter setting from the original papers. We assess sample quality using three metrics: FID (Heusel et al., 2017), CLIP score (Radford et al., 2021), and ImageReward (Xu et al., 2024). ImageReward is designed to align with human preferences, evaluating images based on text alignment, aesthetic appeal, and safety considerations. Quantitative results are summarized in Table 1, while Figure 4 provides a qualitative comparison of generated images. To ensure a fair comparison, fine-grained experimental results for the baseline methods are included in the appendix. The "Best performance" column in Table 1 aggregates these results.

Our experiments reveal several key advantages of ADG. First, ADG demonstrates robustness at high guidance

weights, maintaining stable performance even under conditions where baselines exhibit significant color distortions and quality degradation. Second, ADG consistently outperforms baselines across all guidance weights in text-image consistency metrics (CLIP and IR) while remaining competitive in FID. Notably, ADG achieves substantial improvements in IR, indicating better alignment with human preferences. Finally, ADG exhibits a wide operational range, delivering superior generation performance across a broad spectrum of guidance weights. In contrast, baseline methods experience rapid performance decay beyond their optimal operating points, highlighting the limitations of these methods in comparison to ADG.

## 5.2. Ablation Study

We investigate two variants of the proposed ADG algorithm, described in Appendix E as Algorithm 6 and Algorithm 7. The first variant removes the maximum turning angle constraint, while the second introduces normalization to the norm of $\hat{x}_{0,\omega}$. Table 2 presents the performance metrics for these variants under $\omega = 8$.

*Table 2.* Results of 10 NFE generation with SD v3.5 (d=38) on COCO10k under $\omega = 8$.

| Method | CLIP ↑ | IR ↑ | FID ↓ |
|---|---|---|---|
| ADG (Proposed) | **0.322** | **0.970** | 16.7 |
| ADG w/o angle constraint | 0.275 | -0.782 | 28.6 |
| ADG w normalization | **0.322** | 0.958 | **16.6** |

The experimental results indicate that normalization has a limited impact on ADG's performance. The comparable performance between the normalized ADG and the original framework demonstrates that the ADG algorithm effectively controls the norm of generated samples. This finding further confirms that angular domain adjustment, rather than norm adjustment, serves as the primary mechanism for enhancing

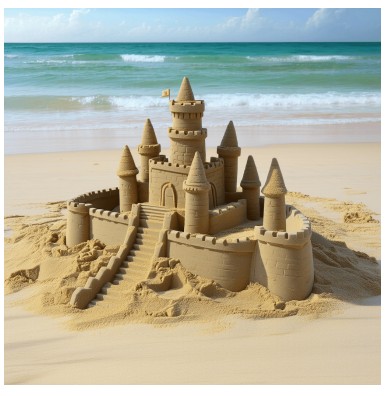 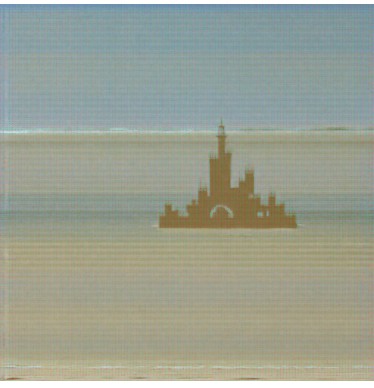 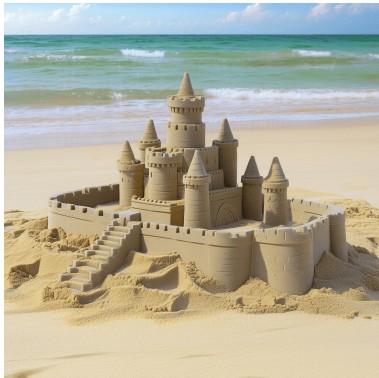

| 6.a ADG | 6.b ADG without angle constraint | 6.c ADG with normalization |

*Figure 6.* Comparison of different ADG variants. ADG without angle constraint exhibits significant degradation, while ADG with normalization remains stable.

text-image alignment.

Furthermore, the necessity of the maximum turning angle constraint is evident. Removing this constraint leads to a substantial decline in performance across all evaluation metrics. As illustrated in Figure 6, the unconstrained variant exhibits severe instability, particularly when the angular deviation $(\omega - 1)\gamma$ urpasses the critical threshold of $\pi$, leading to pathological guidance directions and ultimately catastrophic generation failures.

### 5.3. Compatibility with High order Samplers

To demonstrate the compatibility of our proposed Angular Domain Guidance (ADG) algorithm with different models and samplers, we conducted experiments with SD v2.1 using DPM-Solver sampler(Lu et al., 2022a). As shown in Table 3, ADG consistently outperforms CFG across all evaluation metrics, achieving higher CLIP scores and ImageReward while maintaining competitive FID performance. These results highlight ADG's seamless integration with high-order samplers and its ability to enhance performance across multiple dimensions.

*Table 3.* Results of 25 NFE generation with SD v2.1 on COCO 10k using DPM-Solver under $\omega = 8$.

| METHOD | CLIP ↑ | IR ↑ | FID ↓ |
|---|---|---|---|
| ADG (PROPOSED) | **0.322** | **0.467** | **16.6** |
| CFG | 0.313 | 0.399 | **16.6** |

## 6. Discussion

Angle Domain Guidance (ADG) is introduced as an alternative to the widely adopted Classifier-Free Guidance (CFG) in latent-diffusion-based text-to-image (T2I) genera-

tion. ADG addresses critical limitations of CFG, including norm amplification and oversaturation under high guidance weights, by leveraging angular-domain updates in the latent space. This heuristic approach stabilizes the sampling process and consistently improves semantic alignment, image fidelity, and diversity metrics.

The theoretical analysis presented in this work primarily focuses on the limitations of CFG. Specifically, we demonstrate how linear-domain amplification in CFG leads to deviations from the target distribution, resulting in undesirable artifacts such as oversaturation. While these insights informed the design of ADG, it is important to note that, like CFG, ADG remains a heuristic approach without formal theoretical guarantees.

In comparison to existing methods, ADG introduces a novel perspective by shifting the focus from linear to angular-domain adjustments. This shift not only mitigates the identified issues with CFG but also ensures seamless integration with current samplers and latent diffusion frameworks. However, ADG is specifically designed for latent diffusion models, which dominate modern T2I generation due to their computational efficiency. If future advancements explore diffusion processes in the image domain, ADG may require significant adaptations to remain effective.

The ADG framework is highly extensible and can be adapted to other generative tasks, such as video synthesis(Liu et al., 2024), 3D content generation(Lin et al., 2023) or depth impainting(Sun et al., 2025). Future research could also explore integrating ADG with techniques aimed at reducing sampling iterations(Li & Cai, 2024; Lu et al., 2022a), to enhance its efficiency for real-time applications. These extensions would further solidify ADG's role as a versatile and robust solution for advancing generative modeling.

# 7. Conclusion

We present Angular Domain Guidance (ADG), a novel approach designed to address the limitations of Classifier-Free Guidance (CFG) in diffusion-based text-to-image generation. By leveraging angular-domain updates in the latent space, ADG enable stable and high-quality image generation. Experimental results demonstrate that ADG consistently outperforms existing methods across multiple evaluation metrics, including semantic alignment, image fidelity, and diversity. These findings underscore ADG's potential as a robust and effective alternative for advancing text-to-image generation.

## Acknowledgements

This work was supported by NSAF (Grant No. U2230201) and a Grant from the Guoqiang Institute, Tsinghua University. The authors are affiliated with the Department of Electronic Engineering and the Beijing National Research Center for Information Science and Technology at Tsinghua University.

## Impact Statement

This work re-examines the foundational mechanisms of guidance in latent diffusion models and presents a principled alternative to the widely used Classifier-Free Guidance (CFG). By conducting a theoretical analysis of CFG's failure modes—particularly norm amplification and anomalous diffusion under high guidance weights—the paper identifies key limitations in linear-domain extrapolation. The proposed Angle Domain Guidance (ADG) algorithm addresses these issues through angular-domain updates, offering a conceptually novel and practically effective framework. By shifting guidance from algebraic extrapolation to geometric rotation, ADG provides a new lens for understanding and designing structure-preserving sampling strategies in generative modeling.

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

# A. Related Work

## A.1. Theoretical Analyses of Classifier-Free Guidance

To the best of our knowledge, four key studies have provided theoretical analyses of Classifier-Free Guidance (CFG). Below, we elucidate the connections and distinctions between our theoretical contributions and these prior works.

**Comparison to Chidambaram et al.(Chidambaram et al., 2024).**

This work studies the performance of the CFG-based ODE sampling method in two scenarios: (1) one-dimensional compactly supported distributions with two components, and (2) one-dimensional Gaussian mixture distributions with two components. They conclude that as the guidance weight increases, the sampling process tends to concentrate on the edges of the conditional distribution. Additionally, even a small nonzero error in score estimation can cause sampling results to deviate significantly from the target distribution's support under sufficiently large guidance weights.

The key difference in our theoretical analysis lies in its broader applicability to high-dimensional settings with multiple components. We demonstrate that the phenomenon of sampling concentrating on the edges only occurs for surface classes, a distinction that is absent in (Chidambaram et al., 2024) due to the nature of their one-dimensional two-component scenarios, where all classes inherently behave as surface classes. By incorporating this insight, our work provides a more nuanced understanding of CFG's behavior in complex, high-dimensional generative tasks, offering theoretical explanations for phenomena that remain unaddressed in their framework.

**Comparison to Wu et al.(Wu et al., 2024).**

This work investigates the impact of the guidance weight on sampling performance under Gaussian Mixture Models (GMMs). The authors evaluate two key metrics: "classification confidence," characterized by the conditional likelihood of the output distribution, and "distribution diversity," quantified by the differential entropy of the output distribution. They theoretically prove that CFG sampling not only increases classification confidence but also reduces distribution diversity.

Our work differs from theirs in two significant aspects. First, we operate under more relaxed assumptions; while (Wu et al., 2024) requires the components of the Gaussian mixture to be approximately orthogonal, our analysis accommodates more general configurations of GMMs. Second, we argue that high classification confidence is not always desirable. For example, in a one-dimensional GMM with two components centered at 1 and -1, respectively, and sharing the same variance 1, a sampling point located at 1000 exhibit extremely high classification confidence but would not constitute a meaningful sample from the target distribution.

**Comparison to Bradley and Nakkiran(Bradley & Nakkiran, 2024).**

This work analyzes CFG under the assumption that both the conditional and unconditional distributions follow zero-mean one-dimensional Gaussian distributions. Under this assumption, they derive a closed-form solution for the output distribution, showing that it does not correspond to the expected gamma-weighted distribution. Additionally, they conduct numerical studies on one-dimensional two-component Gaussian mixture models, finding that the output distribution similarly fails to match the expected gamma-weighted distribution.

The authors further reinterpret CFG as a predictor-corrector method, alternating between a denoising predictor (based on the ODE of the conditional distribution) and a sharpening corrector (employing Langevin dynamics). This perspective provides new insights into the iterative nature of CFG sampling.

In contrast, our work extends beyond the limitations of one-dimensional Gaussian assumptions. We study CFG in high-dimensional settings involving multi-component distributions and introduce the notion of surface classes. Our analysis reveals that phenomena such as norm amplification are closely linked to the geometry of these surface classes, providing a more general and nuanced understanding of CFG behavior in complex generative settings.

**Comparison to Xia et al.(Xia et al., 2024).**

This work builds on the findings of (Bradley & Nakkiran, 2024) and extends the data assumptions to high-dimensional isotropic Gaussian distributions with different parameters for the conditional and unconditional distributions. Under these assumptions, the authors derive a closed-form solution for the CFG output distribution and further confirm its inconsistency with the gamma-weighted distribution. Additionally, they propose relaxing the constraints in the gamma-weighted distribution by introducing more flexible guidance coefficients, allowing the corrected distribution to better align with diffusion theory.

In contrast, we abandon the Gaussian distribution assumption to gain deeper insights into the behavior of CFG output distributions in more general scenarios. Our analysis covers high-dimensional settings with multiple components and highlights the geometric and probabilistic distinctions that emerge in such cases. Specifically, we introduce the concept of surface classes, demonstrating that the phenomena of norm amplification and edge sampling predominantly occur for surface classes. By broadening the theoretical scope and proposing practical solutions such as angular-domain adjustments, our work addresses the limitations of CFG in real-world generative tasks with greater generality and flexibility.

### A.2. Improvements to Classifier-Free Guidance

**Comparison to CFG++(Chung et al., 2025).**

This work, inspired by diffusion model-based inverse problem solvers, proposes performing denoising under the guidance of a weight smaller than 1, followed by re-noising guided by the unconditional distribution. However, this method is equivalent to using a time-varying guidance weight in CFG, defined as $\omega_t = \lambda \frac{\sqrt{\bar{\beta}_t \bar{\alpha}_{t-1}}}{\sqrt{\bar{\beta}_t \bar{\alpha}_{t-1}} - \sqrt{\bar{\beta}_{t-1} \bar{\alpha}_t}}$. CFG++ remains confined to the framework of linear-domain guidance and still exhibits color distortions under high $\lambda$. Additionally, our experiments reveal that its optimal guidance weight is highly sensitive to the inference step size, further limiting its applicability (see Appendix G.5).

---

**Algorithm 2** CFG++

> **Require:** $\boldsymbol{x}_T \sim \mathcal{N}(0, \mathbf{I})$,$0 < \lambda \leq 1$
> **for** $t = T$ **to** $1$ **do**
> $\quad \boldsymbol{\epsilon}_{t,\omega}^{(\boldsymbol{c})} = (1 - \lambda)\epsilon_\theta(\boldsymbol{x}, t, \emptyset) + \lambda\boldsymbol{\epsilon}_\theta(\boldsymbol{x}, t, \boldsymbol{c})$
> $\quad \hat{\boldsymbol{x}}_0^{(\boldsymbol{c})} = (\boldsymbol{x}_t - \sqrt{1 - \bar{\alpha}_t}\boldsymbol{\epsilon}_{t,\omega}^{(\boldsymbol{c})})/\sqrt{\bar{\alpha}_t}$
> $\quad \boldsymbol{x}_{t-1} = \sqrt{\bar{\alpha}_{t-1}}\hat{\boldsymbol{x}}_0^{(\boldsymbol{c})} + \sqrt{1 - \bar{\alpha}_{t-1}}\epsilon_\theta(\boldsymbol{x}, t, \emptyset)$
> **end for**

---

**Comparison to PCG(Bradley & Nakkiran, 2024).**

This work decomposes each iteration of the DDPM-style CFG sampler into a denoising step, corresponding to one step of DDIM, and a sharpening step, corresponding to one step of Langevin dynamics. The proposed PCG algorithm performs one denoising step followed by $N$ sharpening steps within each iteration, aiming to enhance the output. However, this enhancement is also carried out within the linear domain. Due to the $N$-fold increase in the number of function evaluations (NFE) compared to standard CFG algorithms, its practicality is significantly reduced. Furthermore, the authors explicitly state, "we do present PCG primarily as a tool to understand CFG." As such, we did not compare the proposed algorithm in this work with our method.

---

**Algorithm 3** PCG

> **Require:** $\boldsymbol{x}_T \sim \mathcal{N}(0, \mathbf{I})$,$1 < \omega \in \mathbb{R}$
> **for** $t = T$ **to** $1$ **do**
> $\quad \boldsymbol{s} = -\bar{\beta}_t \epsilon_\theta(\boldsymbol{x}_t, t, \boldsymbol{c})$
> $\quad \hat{\boldsymbol{x}}_0^{(\boldsymbol{c})} = (\boldsymbol{x}_t - \sqrt{1 - \bar{\alpha}_t}\epsilon_\theta(\boldsymbol{x}_t, t, \boldsymbol{c}))/\sqrt{\bar{\alpha}_t}$
> $\quad \boldsymbol{x}_{t-1} = \sqrt{\bar{\alpha}_{t-1}}\hat{\boldsymbol{x}}_0^{(\boldsymbol{c})} + \sqrt{1 - \bar{\alpha}_{t-1}}\frac{\boldsymbol{x}_t - \hat{\boldsymbol{x}}_0^{(\boldsymbol{c})}}{\sqrt{1 - \bar{\alpha}_t}}$
> $\quad \kappa = \left(1 - \frac{\bar{\alpha}_t}{\bar{\alpha}_{t-1}}\right)$
> $\quad$ **for** $k = 1$ **to** $N$ **do**
> $\quad\quad \eta \sim \mathcal{N}(0, \mathbf{I})$
> $\quad\quad \boldsymbol{\epsilon}_{t-1,\omega}^{(\boldsymbol{c})} = (1 - \omega)\epsilon_\theta(\boldsymbol{x}, t-1, \emptyset) + \omega\boldsymbol{\epsilon}_\theta(\boldsymbol{x}, t-1, \boldsymbol{c})$
> $\quad\quad \boldsymbol{x}_{t-1} = \boldsymbol{x}_{t-1} - \frac{\kappa}{2}\frac{\boldsymbol{\epsilon}_{t-1,\omega}^{(\boldsymbol{c})}}{\bar{\beta}_{t-1}} + \sqrt{\kappa}\eta$
> $\quad$ **end for**
> **end for**

---

**Comparison to ReCFG(Xia et al., 2024).**

This work relaxes the constraint of the weighting coefficients summing to one in traditional CFG by introducing a precomputed lookup table $f$. Strict implementation of ReCFG requires precomputing this lookup table for all conditions, which is impractical for open-condition models like T2I. For datasets like ImageNet, where images have explicit category labels, the authors propose precomputing the table based on categories rather than text conditions. However, in real-world scenarios, text prompts do not always correspond to specific categories, making this method unsuitable for general T2I tasks. Consequently, we do not compare their algorithm with ours.

---

**Algorithm 4** ReCFG

---

**Require:** $x_T \sim \mathcal{N}(0, \mathbf{I}), 1 < \omega \in \mathbb{R}$, trained lookup table $f$.
**for** $t = T$ **to** 1 **do**
  $\lambda = f(c)$
  $\epsilon_{t,\omega}^{(c)} = \lambda(1-\omega)\epsilon_\theta(x, t, \emptyset) + \omega\epsilon_\theta(x, t, c)$
  $\hat{x}_0^{(c)} = (x_t - \sqrt{1-\bar{\alpha}_t}\epsilon_{t,\omega}^{(c)})/\sqrt{\bar{\alpha}_t}$
  $x_{t-1} = \sqrt{\bar{\alpha}_{t-1}}\hat{x}_{0,\omega} + \sqrt{1-\bar{\alpha}_{t-1}}\epsilon_{t,\omega}^{(c)}$
**end for**

---

**Comparison to APG(Sadat et al., 2025)**

This work attributes image oversaturation and degradation to the parallel component of the difference vector $\Delta\hat{x}_0 = \hat{x}_0^{(c)} - \hat{x}_0^{(\emptyset)}$ with respect to $\hat{x}_0^{(c)}$, denoted as $\Delta\hat{x}_{0,\parallel}$. Building on this observation, they propose reducing the influence of the parallel component in CFG by replacing $\Delta\hat{x}_{0,CFG} = \Delta\hat{x}_{0,\parallel} + \Delta\hat{x}_{0,\perp}$ with $\Delta\hat{x}_{0,APG} = \eta\Delta\hat{x}_{0,\parallel} + \Delta\hat{x}_{0,\perp}$, where $\eta < 1$, to mitigate its adverse effect on image quality. In addition, this work introduces an extra negative momentum mechanism and imposes a constraint on the norm of $\Delta\hat{x}_0$. While removing the parallel component slows the norm growth, it does not fully resolve the issue of norm amplification. However, its algorithm does not directly constrain the norm of $\Delta\hat{x}_{0,CFG}$, and thus fails to address the issue of excessive latent norm under high guidance weights. As a result, image degradation still occurs when the guidance weight is large.

---

**Algorithm 5** APG

---

**Require:** $x_T \sim \mathcal{N}(0, \mathbf{I}), 1 < \omega \in \mathbb{R}, \beta < 0, r \in \mathbb{R}_+, 0 \leq \eta < 1$
$\Delta\hat{x}_{0,history} = \mathbf{0}$
**for** $t = T$ **to** 1 **do**
  $\hat{x}_0^{(c)} = (x_t - \sqrt{1-\bar{\alpha}_t}\epsilon_\theta(x, t, c))/\sqrt{\bar{\alpha}_t}$
  $\hat{x}_0^{(\emptyset)} = (x_t - \sqrt{1-\bar{\alpha}_t}\epsilon_\theta(x, t, \emptyset))/\sqrt{\bar{\alpha}_t}$
  $\Delta\hat{x}_0 = \hat{x}_0^{(c)} - \hat{x}_0^{(\emptyset)}$
  $\Delta\hat{x}_0^{\parallel} = \frac{\langle\Delta\hat{x}_0, \hat{x}_0^{(c)}\rangle}{\langle\hat{x}_0^{(c)}, \hat{x}_0^{(c)}\rangle}\hat{x}_0^{(c)}$
  $\Delta\hat{x}_0^{\perp} = \Delta\hat{x}_0 - \Delta\hat{x}_0^{\parallel}$
  $\Delta\hat{x}_0 = \eta\Delta\hat{x}_0^{\parallel} + \Delta\hat{x}_0^{\perp}$
  $\Delta\hat{x}_0 = \Delta\hat{x}_0 \min\left(1, \frac{r}{\|\Delta\hat{x}_0\|}\right)$
  $\Delta\hat{x}_{0,history} = \Delta\hat{x}_0 - \beta\Delta\hat{x}_{0,history}$
  $\hat{x}_{0,\omega} = \hat{x}_0^{(c)} + (\omega - 1)\Delta\hat{x}_{0,history}$
  $x_{t-1} = \sqrt{\bar{\alpha}_{t-1}}\hat{x}_{0,\omega} + \sqrt{1-\bar{\alpha}_{t-1}}\frac{x_t - \bar{\alpha}_t\hat{x}_{0,\omega}}{\sqrt{1-\bar{\alpha}_t}}$
**end for**

---

## B. Proof of Theorem 3.2

We now consider a strengthened proposition of Theorem 3.2:

**Theorem B.1.** *Assume the data follows the model* (11) *and* $c^*$ *is the surface class of the distribution. Consider two state*

variables $\boldsymbol{x}_t$ and $\boldsymbol{z}_t$ governed by the following ODEs:

$$\frac{\mathrm{d}\boldsymbol{x}_t}{\mathrm{d}t} = \frac{\beta(t)}{2}\left[-\boldsymbol{x}_t - \nabla_{\boldsymbol{x}_t}\log p_t(\boldsymbol{x}_t|c^*)\right], \tag{20}$$

$$\frac{\mathrm{d}\boldsymbol{z}_t}{\mathrm{d}t} = \frac{\beta(t)}{2}\left[-\boldsymbol{z}_t - (1+\omega)\nabla_{\boldsymbol{z}_t}\log p_t(\boldsymbol{z}_t|c^*) + \omega\nabla_{\boldsymbol{z}_t}\log p_t(\boldsymbol{z}_t)\right]. \tag{21}$$

If the initial conditions satisfy $\boldsymbol{x}_T^\top\boldsymbol{\mu}_{c^*} \leq \boldsymbol{z}_T^\top\boldsymbol{\mu}_{c^*}$, then for any $t \in [0,T)$, the following inequality holds:

$$\boldsymbol{x}_t^\top\boldsymbol{\mu}_{c^*} < \boldsymbol{z}_t^\top\boldsymbol{\mu}_{c^*}.$$

*Proof.* For Gaussian distributions and Gaussian mixture distributions, the score functions admit closed-form expressions. For the conditional score function:

$$\nabla_{\boldsymbol{x}}\log p_t(\boldsymbol{x}|c^*) = \nabla_{\boldsymbol{x}}\left(-\frac{d}{2}\log(2\pi) - \frac{\|\boldsymbol{x} - \bar{\alpha}_t\boldsymbol{\mu}_{c^*}\|_2^2}{2}\right)$$

$$= -\boldsymbol{x} + \bar{\alpha}_t\boldsymbol{\mu}_{c^*}, \tag{22}$$

and for the marginal score function:

$$\nabla_{\boldsymbol{x}}\log p_t(\boldsymbol{x}) = \frac{\nabla_{\boldsymbol{x}}\left(\sum_{c=1}^C \pi_c\mathcal{N}(\boldsymbol{x}|\bar{\alpha}_t\boldsymbol{\mu}_c, \mathbf{I})\right)}{\sum_{c=1}^C \pi_c\mathcal{N}(\boldsymbol{x}|\bar{\alpha}_t\boldsymbol{\mu}_c, \mathbf{I})}$$

$$= -\boldsymbol{x} + \bar{\alpha}_t\sum_{c=1}^C \pi_c^*(\boldsymbol{x})\boldsymbol{\mu}_c, \tag{23}$$

where $\pi_c^*(\boldsymbol{x}) = \frac{\pi_c\mathcal{N}(\boldsymbol{x}|\bar{\alpha}_t\boldsymbol{\mu}_c, \mathbf{I})}{\sum_{c'=1}^C \pi_{c'}\mathcal{N}(\boldsymbol{x}|\bar{\alpha}_t\boldsymbol{\mu}_{c'}, \mathbf{I})}$.

Substituting (22) and (23) into (20) and (21) yields:

$$\frac{\mathrm{d}\boldsymbol{x}_t}{\mathrm{d}t} = \frac{\beta(t)}{2}\left[-\bar{\alpha}_t\boldsymbol{\mu}_{c^*}\right], \tag{24}$$

$$\frac{\mathrm{d}\boldsymbol{z}_t}{\mathrm{d}t} = \frac{\beta(t)}{2}\left[-\bar{\alpha}_t\boldsymbol{\mu}_{c^*} - \omega\bar{\alpha}_t\left(\boldsymbol{\mu}_{c^*} - \sum_{c=1}^C \pi_c^*(\boldsymbol{x})\boldsymbol{\mu}_c\right)\right]. \tag{25}$$

By taking the projection along $\boldsymbol{w}^\top\boldsymbol{\mu}_{c^*}$, we have:

$$\frac{\mathrm{d}\boldsymbol{w}^\top\boldsymbol{x}_t}{\mathrm{d}t} = \frac{\beta(t)}{2}\left[-\bar{\alpha}_t\boldsymbol{w}^\top\boldsymbol{\mu}_{c^*}\right], \tag{26}$$

$$\frac{\mathrm{d}\boldsymbol{w}^\top\boldsymbol{z}_t}{\mathrm{d}t} = \frac{\beta(t)}{2}\left[-\bar{\alpha}_t\boldsymbol{w}^\top\boldsymbol{\mu}_{c^*} - \omega\bar{\alpha}_t\sum_{c=1}^C \pi_c^*(\boldsymbol{x})\left(\boldsymbol{w}^\top\boldsymbol{\mu}_{c^*} - \boldsymbol{w}^\top\boldsymbol{\mu}_c\right)\right]. \tag{27}$$

From Definition 3.1, it follows that:

$$\boldsymbol{w}^\top\boldsymbol{\mu}_{c^*} - \boldsymbol{w}^\top\boldsymbol{\mu}_c < 0, \quad \forall c \neq c^*,$$

which implies:

$$\frac{\mathrm{d}\boldsymbol{w}^\top\boldsymbol{z}_t}{\mathrm{d}t} < \frac{\mathrm{d}\boldsymbol{w}^\top\boldsymbol{x}_t}{\mathrm{d}t}, \quad \forall t \in [0,T).$$

Using the ODE comparison theorem and the initial condition $\boldsymbol{x}_T^\top\boldsymbol{\mu}_{c^*} \leq \boldsymbol{z}_T^\top\boldsymbol{\mu}_{c^*}$ (notice that the ODE is time-reversed), we conclude that:

$$\boldsymbol{x}_t^\top\boldsymbol{\mu}_{c^*} < \boldsymbol{z}_t^\top\boldsymbol{\mu}_{c^*}, \quad \forall t \in [0,T).$$

$\square$

# C. proof of Theorem 3.3

Using Lemma 3.4, we express the score functions as:

$$\nabla_{\boldsymbol{x}} \log p_t(\boldsymbol{x}|c^*) = \frac{\sqrt{\bar{\alpha}_t}\mathbb{E}_{\boldsymbol{x}_0 \sim q_{c^*,t,\boldsymbol{x}}}[\boldsymbol{x}_0] - \boldsymbol{x}}{\bar{\beta}_t}, \tag{28}$$

$$\nabla_{\boldsymbol{x}} \log p_t(\boldsymbol{x}|\emptyset) = \frac{\sqrt{\bar{\alpha}_t}\mathbb{E}_{\boldsymbol{x}_0 \sim q_{\emptyset,t,\boldsymbol{x}}}[\boldsymbol{x}_0] - \boldsymbol{x}}{\bar{\beta}_t}, \tag{29}$$

where the conditional distributions $q_{c^*,t,\boldsymbol{x}}$ and $q_{\emptyset,t,\boldsymbol{x}}$ are given by:

$$q_{c^*,t,\boldsymbol{x}}(\boldsymbol{x}_0) \propto \mathcal{N}(\boldsymbol{x}_0; \boldsymbol{\mu}_{c^*}, \mathbf{I}) \exp\left(-\frac{\|\boldsymbol{x} - \sqrt{\bar{\alpha}_t}\boldsymbol{x}_0\|_2^2}{2\bar{\beta}_t}\right), \tag{30}$$

$$q_{\emptyset,t,\boldsymbol{x}}(\boldsymbol{x}_0) \propto \sum_{c=1}^{C} w_c \mathcal{N}(\boldsymbol{x}_0; \boldsymbol{\mu}_c, \mathbf{I}) \exp\left(-\frac{\|\boldsymbol{x} - \sqrt{\bar{\alpha}_t}\boldsymbol{x}_0\|_2^2}{2\bar{\beta}_t}\right). \tag{31}$$

For $q_{c^*,t,\boldsymbol{x}}$, we calculate:

$$\mathcal{N}(\boldsymbol{x}_0; \boldsymbol{\mu}_{c^*}, \mathbf{I}) \exp\left(-\frac{\|\boldsymbol{x} - \sqrt{\bar{\alpha}_t}\boldsymbol{x}_0\|_2^2}{2\bar{\beta}_t}\right) \propto \exp\left(-\|\boldsymbol{x}_0 - \boldsymbol{\mu}_{c^*}\|_2^2 - \frac{\|\boldsymbol{x} - \sqrt{\bar{\alpha}_t}\boldsymbol{x}_0\|_2^2}{2\bar{\beta}_t}\right) \tag{32}$$

$$\propto \exp\left(-\left(1 + \frac{\bar{\alpha}_t}{\bar{\beta}_t}\right)\boldsymbol{x}_0^\top \boldsymbol{x}_0 + 2\left(\boldsymbol{\mu}_{c^*} + \frac{\sqrt{\bar{\alpha}_t}}{\bar{\beta}_t}\boldsymbol{x}\right)^\top \boldsymbol{x}_0\right). \tag{33}$$

Since $\bar{\beta}_t = 1 - e^{-2t}$ and $\bar{\alpha}_t = e^{-2t}$, the expectation is:

$$\mathbb{E}_{\boldsymbol{x}_0 \sim q_{c^*,t,\boldsymbol{x}}}[\boldsymbol{x}_0] = \bar{\beta}_t \boldsymbol{\mu}_{c^*} + \sqrt{\bar{\alpha}_t}\boldsymbol{x}. \tag{34}$$

Substituting this back, we find:

$$\nabla_{\boldsymbol{x}} \log p_t(\boldsymbol{x}|c^*) = \frac{\sqrt{\bar{\alpha}_t}(\bar{\beta}_t \boldsymbol{\mu}_{c^*} + \sqrt{\bar{\alpha}_t}\boldsymbol{x}) - \boldsymbol{x}}{\bar{\beta}_t} \tag{35}$$

$$= \sqrt{\bar{\alpha}_t}\boldsymbol{\mu}_{c^*} - \boldsymbol{x}. \tag{36}$$

Thus, for $\boldsymbol{x} = \sqrt{\bar{\alpha}_t}\boldsymbol{\mu}_{c^*} + k\boldsymbol{w}$, we have:

$$\nabla_{\boldsymbol{x}} \log p_t(\boldsymbol{x}|c^*) = -k\boldsymbol{w}. \tag{37}$$

similarly, we have

$$\sum_{c=1}^{C} w_c \mathcal{N}(\boldsymbol{x}_0; \boldsymbol{\mu}_c, \mathbf{I}) \exp\left(-\frac{\|\boldsymbol{x} - \sqrt{\bar{\alpha}_t}\boldsymbol{x}_0\|_2^2}{2\bar{\beta}_t}\right) \propto \sum_{c=1}^{C} w_c \exp\left(-\|\boldsymbol{x}_0 - \boldsymbol{\mu}_c\|_2^2 - \frac{\|\boldsymbol{x} - \sqrt{\bar{\alpha}_t}\boldsymbol{x}_0\|_2^2}{2\bar{\beta}_t}\right) \tag{38}$$

$$\propto \sum_{c=1}^{C} w_{c,\boldsymbol{x}}^{(1)} \mathcal{N}\left(\boldsymbol{x}_0; \bar{\beta}_t \boldsymbol{\mu}_c + \sqrt{\bar{\alpha}_t}\boldsymbol{x}, \frac{1}{\sqrt{\bar{\beta}_t}}\mathbf{I}\right), \tag{39}$$

where

$$w_{c,\boldsymbol{x}}^{(1)} = w_c \exp\left(\|\boldsymbol{x} - \sqrt{\bar{\alpha}_t}\boldsymbol{\mu}_c\|_2^2\right). \tag{40}$$

Let

$$w_{c,\boldsymbol{x}}^{(3)} = \frac{w_c \mathcal{N}(\boldsymbol{x}; \sqrt{\bar{\alpha}_t}\boldsymbol{\mu}_c, \mathbf{I})}{\sum_{l=1}^{C} w_l \mathcal{N}(\boldsymbol{x}; \sqrt{\bar{\alpha}_t}\boldsymbol{\mu}_l, \mathbf{I})}.$$

We then have:

$$q_{\emptyset,t,\boldsymbol{x}}(\boldsymbol{x}_0) = \sum_{c=1}^{C} w_{c,\boldsymbol{x}}^{(3)} \mathcal{N}\left(\boldsymbol{x}_0; \bar{\beta}_t \boldsymbol{\mu}_c + \sqrt{\bar{\alpha}_t}\boldsymbol{x}, \frac{1}{\sqrt{\bar{\beta}_t}}\mathbf{I}\right). \tag{41}$$

From this, the expectation of $\boldsymbol{x}_0$ with respect to $q_{\emptyset,t,\boldsymbol{x}}$ becomes:

$$\mathbb{E}_{\boldsymbol{x}_0 \sim q_{\emptyset,t,\boldsymbol{x}}}[\boldsymbol{x}_0] = \sum_{c=1}^{C} w_{c,\boldsymbol{x}}^{(3)} \left(\bar{\beta}_t \boldsymbol{\mu}_c + \sqrt{\bar{\alpha}_t}\boldsymbol{x}\right)$$

$$= \sqrt{\bar{\alpha}_t}\boldsymbol{x} + \bar{\beta}_t \sum_{c=1}^{C} w_{c,\boldsymbol{x}}^{(3)} \boldsymbol{\mu}_c. \tag{42}$$

Now consider the difference between the expectations under $q_{c^*,t,\boldsymbol{x}}$ and $q_{\emptyset,t,\boldsymbol{x}}$:

$$\mathbb{E}_{\boldsymbol{x}_0 \sim q_{c^*,t,\boldsymbol{x}}}[\boldsymbol{x}_0] - \mathbb{E}_{\boldsymbol{x}_0 \sim q_{\emptyset,t,\boldsymbol{x}}}[\boldsymbol{x}_0] = \bar{\beta}_t \sum_{c=1}^{C} w_{c,\boldsymbol{x}}^{(3)} \left(\boldsymbol{\mu}_{c^*} - \boldsymbol{\mu}_c\right)$$

$$= \bar{\beta}_t \sum_{c \neq c^*} w_{c,\boldsymbol{x}}^{(3)} \left(\boldsymbol{\mu}_{c^*} - \boldsymbol{\mu}_c\right), \tag{43}$$

where the second equality uses the fact that for $c = c^*$, the term cancels out.

Next, we calculate the dot product of $\boldsymbol{s}_{cfg,\omega}$ with $\nabla \log p_t(\boldsymbol{x}|c^*)$:

$$\boldsymbol{s}_{cfg,\omega}^{\top} \nabla \log p_t(\boldsymbol{x}|c^*) = \left((\omega - 1)\frac{\sqrt{\bar{\alpha}_t}\left(\mathbb{E}_{\boldsymbol{x}_0 \sim q_{c^*,t,\boldsymbol{x}}}[\boldsymbol{x}_0] - \mathbb{E}_{\boldsymbol{x}_0 \sim q_{\emptyset,t,\boldsymbol{x}}}[\boldsymbol{x}_0]\right)}{\bar{\beta}_t} + \nabla \log p_t(\boldsymbol{x}|c^*)\right)^{\top} \nabla \log p_t(\boldsymbol{x}|c^*)$$

$$= -k(\omega - 1)\frac{\sqrt{\bar{\alpha}_t}}{\bar{\beta}_t} \sum_{c \neq c^*} w_{c,\boldsymbol{x}}^{(3)} \left(\boldsymbol{\mu}_{c^*} - \boldsymbol{\mu}_c\right)^{\top} \boldsymbol{w} + k^2 \boldsymbol{w}^{\top}\boldsymbol{w}, \tag{44}$$

where the second term $\nabla \log p_t(\boldsymbol{x}|c^*)$ contributes $k^2 \boldsymbol{w}^{\top}\boldsymbol{w}$ and the projection of the difference in expectations contributes the first term.

Without loss of generality, assume $\boldsymbol{w}$ is a unit vector. From Definition 3.1, we know that $\boldsymbol{\mu}_{c^*}^{\top}\boldsymbol{w} > \boldsymbol{\mu}_c^{\top}\boldsymbol{w}$ for $c \neq c^*$. Define the following terms for clarity:

$$\delta = \min_{c \neq c^*}(\boldsymbol{\mu}_{c^*} - \boldsymbol{\mu}_c)^{\top}\boldsymbol{w} > 0, \tag{45}$$

$$R^2 = \max_{c \in c^*}\|\boldsymbol{\mu}_c\|_2^2, \tag{46}$$

$$\lambda = \min_{c \neq c^*}\frac{w_c}{w_{c^*}}. \tag{47}$$

Now consider the dot product of $\boldsymbol{s}_{cfg,\omega}$ and $\nabla \log p_t(\boldsymbol{x}|c^*)$. For $k < \sqrt{\bar{\alpha}_t}R$, we have:

$$\boldsymbol{s}_{cfg,\omega}^{\top} \nabla \log p_t(\boldsymbol{x}, c^*) \leq -k(\omega - 1)\frac{\sqrt{\bar{\alpha}_t}}{\bar{\beta}_t} \sum_{c \neq c^*} w_{c,\boldsymbol{x}}^{(3)}\delta + k^2. \tag{48}$$

Substituting the expression for $w_{c,\boldsymbol{x}}^{(3)}$, we get:

$$\boldsymbol{s}_{cfg,\omega}^{\top} \nabla \log p_t(\boldsymbol{x}, c^*) \leq k^2 - k(\omega - 1)\frac{\sqrt{\bar{\alpha}_t}}{\bar{\beta}_t}\delta \sum_{c \neq c^*} \frac{w_c \mathcal{N}(\boldsymbol{x}; \sqrt{\bar{\alpha}_t}\boldsymbol{\mu}_c, \mathbf{I})}{\sum_{l=1}^{C} w_l \mathcal{N}(\boldsymbol{x}; \sqrt{\bar{\alpha}_t}\boldsymbol{\mu}_l, \mathbf{I})}. \tag{49}$$

Using the decomposition of the weights, we write:

$$\sum_{c \neq c^*} \frac{w_c \mathcal{N}(\boldsymbol{x}; \sqrt{\bar{\alpha}_t}\boldsymbol{\mu}_c, \mathbf{I})}{\sum_{l=1}^{C} w_l \mathcal{N}(\boldsymbol{x}; \sqrt{\bar{\alpha}_t}\boldsymbol{\mu}_l, \mathbf{I})} = 1 - \frac{w_{c^*} \mathcal{N}(\boldsymbol{x}; \sqrt{\bar{\alpha}_t}\boldsymbol{\mu}_{c^*}, \mathbf{I})}{\sum_{l=1}^{C} w_l \mathcal{N}(\boldsymbol{x}; \sqrt{\bar{\alpha}_t}\boldsymbol{\mu}_l, \mathbf{I})}. \tag{50}$$

Substitute the Gaussian density terms and simplify:

$$\frac{w_{c^*} \mathcal{N}(\boldsymbol{x}; \sqrt{\bar{\alpha}_t}\boldsymbol{\mu}_{c^*}, \mathbf{I})}{\sum_{l=1}^{C} w_l \mathcal{N}(\boldsymbol{x}; \sqrt{\bar{\alpha}_t}\boldsymbol{\mu}_l, \mathbf{I})} = \frac{1}{1 + \sum_{l \neq c^*} \frac{w_l}{w_{c^*}} \frac{\mathcal{N}(\boldsymbol{x}; \sqrt{\bar{\alpha}_t}\boldsymbol{\mu}_l, \mathbf{I})}{\mathcal{N}(\boldsymbol{x}; \sqrt{\bar{\alpha}_t}\boldsymbol{\mu}_{c^*}, \mathbf{I})}}. \tag{51}$$

Using the exponential decay property of the Gaussian distribution:

$$\frac{\mathcal{N}(\boldsymbol{x}; \sqrt{\bar{\alpha}_t}\boldsymbol{\mu}_l, \mathbf{I})}{\mathcal{N}(\boldsymbol{x}; \sqrt{\bar{\alpha}_t}\boldsymbol{\mu}_{c^*}, \mathbf{I})} \leq \exp\left(-\frac{\|\sqrt{\bar{\alpha}_t}\boldsymbol{\mu}_{c^*} + k\boldsymbol{w} - \sqrt{\bar{\alpha}_t}\boldsymbol{\mu}_l\|_2^2}{2}\right). \tag{52}$$

Combining terms:

$$\frac{w_{c^*} \mathcal{N}(\boldsymbol{x}; \sqrt{\bar{\alpha}_t}\boldsymbol{\mu}_{c^*}, \mathbf{I})}{\sum_{l=1}^{C} w_l \mathcal{N}(\boldsymbol{x}; \sqrt{\bar{\alpha}_t}\boldsymbol{\mu}_l, \mathbf{I})} \geq \frac{1}{1 + (C-1)\lambda \exp\left(-\frac{9\bar{\alpha}_t R^2}{2}\right)}. \tag{53}$$

Substitute back into the original inequality:

$$\boldsymbol{s}_{cfg,\omega}^{\top} \nabla \log p_t(\boldsymbol{x}, c^*) \leq k^2 - k(\omega - 1)\frac{\sqrt{\bar{\alpha}_t}}{\bar{\beta}_t}\delta\left(1 - \frac{1}{1 + (C-1)\lambda \exp\left(-\frac{9\bar{\alpha}_t R^2}{2}\right)}\right). \tag{54}$$

For $0 < k < \min\left(\sqrt{\bar{\alpha}_t}R, (\omega - 1)\frac{\sqrt{\bar{\alpha}_t}}{\bar{\beta}_t}\delta\left(1 - \frac{1}{1 + (C-1)\lambda \exp\left(-\frac{9\bar{\alpha}_t R^2}{2}\right)}\right)\right)$, we conclude:

$$\boldsymbol{s}_{cfg,\omega}^{\top} \nabla \log p_t(\boldsymbol{x}, c^*) < 0.$$

## D. Proof of Proposition 4.1

For convenience, we define the following notation:

$$\hat{\boldsymbol{x}}_{\text{asist}} = \frac{1}{\sin(\gamma)}\left(\hat{\boldsymbol{x}}_0^{(c)} - \text{proj}_{\hat{\boldsymbol{x}}_0^{(\emptyset)}}(\hat{\boldsymbol{x}}_0^{(c)})\right).$$

It follows that:

$$\|\hat{\boldsymbol{x}}_{\text{asist}}\|_2 = \left\|\hat{\boldsymbol{x}}_0^{(c)}\right\|_2.$$

We now compute the norm of $\hat{\boldsymbol{x}}_{0,\omega}$:

$$\|\hat{\boldsymbol{x}}_{0,\omega}\|_2 = \sqrt{\sin^2\left((\omega-1)\gamma\right)\|\hat{\boldsymbol{x}}_{\text{asist}}\|_2^2 + 2\sin\left((\omega-1)\gamma\right)\cos\left((\omega-1)\gamma\right)\langle\hat{\boldsymbol{x}}_{\text{asist}}, \hat{\boldsymbol{x}}_0^{(c)}\rangle + \cos^2\left((\omega-1)\gamma\right)\left\|\hat{\boldsymbol{x}}_0^{(c)}\right\|_2^2}$$

$$= \left\|\hat{\boldsymbol{x}}_0^{(c)}\right\|_2 \sqrt{1 + \sin(2(\omega-1)\gamma)\frac{\langle\hat{\boldsymbol{x}}_{\text{asist}}, \hat{\boldsymbol{x}}_0^{(c)}\rangle}{\left\|\hat{\boldsymbol{x}}_0^{(c)}\right\|_2^2}}$$

$$\leq \left\|\hat{\boldsymbol{x}}_0^{(c)}\right\|_2 \sqrt{1 + \sin(2(\omega-1)\gamma)}$$

$$\leq \sqrt{2}\left\|\hat{\boldsymbol{x}}_0^{(c)}\right\|_2.$$

This completes the proof.

# E. Variants of ADG

The implementation details of the two variant algorithms mentioned in Section 5 are outlined here. Algorithm 6 removes the constraint on the maximum turning angle, allowing for more flexible updates. On the other hand, Algorithm 7 normalizes the corrected $\hat{x}_0$ to ensure consistency in its magnitude.

---

**Algorithm 6** ADG w/o angle constraint

---

**Require:** $x_T \sim \mathcal{N}(0, \mathbf{I}), 1 < \omega \in \mathbb{R}$
**for** $t = T$ **to** $1$ **do**
   $\hat{x}_0^{(c)} = (x_t - \sqrt{1 - \bar{\alpha}_t}\epsilon_\theta(x, t, c))/\sqrt{\bar{\alpha}_t}$
   $\hat{x}_0^{(\emptyset)} = (x_t - \sqrt{1 - \bar{\alpha}_t}\epsilon_\theta(x, t, \emptyset))/\sqrt{\bar{\alpha}_t}$
   $\gamma = \arccos\left(\frac{(\hat{x}_0^{(\emptyset)})^\top \hat{x}_0^{(c)}}{\|\hat{x}_0^{(\emptyset)}\|_2 \|\hat{x}_0^{(c)}\|_2}\right)$
   $\gamma_\omega = (\omega - 1)\gamma$
   $\hat{x}_{0,\omega} = \cos(\gamma_\omega)\hat{x}_0^{(c)} + \frac{\sin(\gamma_\omega)}{\sin(\gamma)}(\hat{x}_0^{(c)} - \text{proj}_{\hat{x}_0^{(\emptyset)}}(\hat{x}_0^{(c)}))$
   $x_{t-1} = \sqrt{\bar{\alpha}_{t-1}}\hat{x}_{0,\omega} + \sqrt{1 - \bar{\alpha}_{t-1}}\frac{x_t - \bar{\alpha}_t \hat{x}_{0,\omega}}{\sqrt{1 - \bar{\alpha}_t}}$
**end for**

---

**Algorithm 7** ADG with Normalization

---

**Require:** $x_T \sim \mathcal{N}(0, \mathbf{I}), 1 < \omega \in \mathbb{R}$
**for** $t = T$ **to** $1$ **do**
   $\hat{x}_0^{(c)} = (x_t - \sqrt{1 - \bar{\alpha}_t}\epsilon_\theta(x, t, c))/\sqrt{\bar{\alpha}_t}$
   $\hat{x}_0^{(\emptyset)} = (x_t - \sqrt{1 - \bar{\alpha}_t}\epsilon_\theta(x, t, \emptyset))/\sqrt{\bar{\alpha}_t}$
   $\gamma = \arccos\left(\frac{(\hat{x}_0^{(\emptyset)})^\top \hat{x}_0^{(c)}}{\|\hat{x}_0^{(\emptyset)}\|_2 \|\hat{x}_0^{(c)}\|_2}\right)$
   $\gamma_\omega = \text{threshold}((\omega - 1)\gamma, \pi/3)$
   $\hat{x}_{0,\omega} = \cos(\gamma_\omega)\hat{x}_0^{(c)} + \frac{\sin(\gamma_\omega)}{\sin(\gamma)}(\hat{x}_0^{(c)} - \text{proj}_{\hat{x}_0^{(\emptyset)}}(\hat{x}_0^{(c)}))$
   $\hat{x}_{0,\omega} = \hat{x}_{0,\omega}\frac{\|\hat{x}_0^{(c)}\|_2}{\|\hat{x}_{0,\omega}\|_2}$
   $x_{t-1} = \sqrt{\bar{\alpha}_{t-1}}\hat{x}_{0,\omega} + \sqrt{1 - \bar{\alpha}_{t-1}}\frac{x_t - \bar{\alpha}_t \hat{x}_{0,\omega}}{\sqrt{1 - \bar{\alpha}_t}}$
**end for**

---

Moreover, considering that ADG introduces relatively substantial modifications compared to CFG, which may hinder its ease of adoption in existing frameworks, we further propose a simplified variant that can be more seamlessly integrated into current pipelines. This simplified method is also derived from our analysis of CFG, and directly mitigates the degradation issue by explicitly constraining the norm of $\hat{x}_0$.

---

**Algorithm 8** Simplified ADG

---

**Require:** $x_T \sim \mathcal{N}(0, \mathbf{I}), 1 < \omega \in \mathbb{R}$
**for** $t = T$ **to** $1$ **do**
   $\hat{x}_0^{(c)} = (x_t - \sqrt{1 - \bar{\alpha}_t}\epsilon_\theta(x, t, c))/\sqrt{\bar{\alpha}_t}$
   $\hat{x}_0^{(\emptyset)} = (x_t - \sqrt{1 - \bar{\alpha}_t}\epsilon_\theta(x, t, \emptyset))/\sqrt{\bar{\alpha}_t}$
   $\hat{x}_{0,CFG} = \hat{x}_0^{(c)} + (\omega - 1)(\hat{x}_0^{(c)} - \hat{x}_0^{(\emptyset)})$
   $\hat{x}_{0,ADG} = \hat{x}_{0,CFG}\frac{\|\hat{x}_0^{(c)}\|}{\|\hat{x}_{0,CFG}\|}$
   $x_{t-1} = \sqrt{\bar{\alpha}_{t-1}}\hat{x}_{0,ADG} + \sqrt{1 - \bar{\alpha}_{t-1}}\frac{x_t - \bar{\alpha}_t \hat{x}_{0,ADG}}{\sqrt{1 - \bar{\alpha}_t}}$
**end for**

---

*Table 4.* Results of 10 NFE generation with SD v3.5 (d=38) on COCO10k (CFG).

| VALUE | $\omega = 2$ | $\omega = 3$ | $\omega = 3.5$ | $\omega = 4$ | $\omega = 4.5$ |
|---|---|---|---|---|---|
| CLIP↑ | 0.316 | 0.317 | **0.318** | **0.318** | **0.318** |
| IR↑ | 0.711 | 0.831 | **0.843** | 0.835 | 0.792 |
| FID↓ | 17.5 | 17.2 | 17.0 | 16.9 | **16.6** |

| VALUE | $\omega = 5$ | $\omega = 6$ | $\omega = 8$ | $\omega = 10$ | BEST |
|---|---|---|---|---|---|
| CLIP↑ | **0.318** | 0.317 | 0.311 | 0.304 | 0.318 |
| IR↑ | 0.735 | 0.586 | 0.219 | -0.142 | 0.843 |
| FID↓ | **16.6** | 17.1 | 17.2 | 17.7 | 16.6 |

## F. Extension of Flow Matching Model for ADG Algorithm

The generative model based on Flow Matching introduces a time-dependent vector field $\mathbf{v}_t = \frac{\mathrm{d}\boldsymbol{x}_t}{\mathrm{d}t}$, which is used to learn the continuous transformation path from a Gaussian distribution $p_0$ to a target distribution $p_1$ and subsequently perform sampling. Notably, this notation differs from that in diffusion models, where the target distribution is denoted as $p_1$. A key concept in the Flow Matching-based model is the conditional probability paths, which are defined as:

$$p(\boldsymbol{x}_t|\boldsymbol{x}_1) = \mathcal{N}(\boldsymbol{x}_t, \mu_t(\boldsymbol{x}_1), \sigma_t^2(\boldsymbol{x}_1)\mathbf{I}).$$

The conditional probability paths most used in generative tasks are expressed as:

$$p(\boldsymbol{x}_t|\boldsymbol{x}_1) = \mathcal{N}(\boldsymbol{x}_t, t\boldsymbol{x}_1, (1 - (1 - \sigma_{\min})t)^2\mathbf{I}).$$

Under this setting, the ideal reference flow is given by:

$$\mathbf{u}_t = \frac{\mathbb{E}_{\boldsymbol{x}_1|\boldsymbol{x}_t}[\boldsymbol{x}_1] - (1 - \sigma_{\min})\boldsymbol{x}_t}{1 - (1 - \sigma_{\min})t}.$$

Thus, the Flow Matching model's Adaptive Directional Guidance (ADG) can be written as:

$$\hat{\boldsymbol{x}}_1^{(c)} = (1 - (1 - \sigma_{\min})t)\mathbf{v}_t^{(c)}(\boldsymbol{x}_t) + (1 - \sigma_{\min})\boldsymbol{x}_t, \tag{55}$$

$$\hat{\boldsymbol{x}}_1^{(\emptyset)} = (1 - (1 - \sigma_{\min})t)\mathbf{v}_t^{(\emptyset)}(\boldsymbol{x}_t) + (1 - \sigma_{\min})\boldsymbol{x}_t, \tag{56}$$

$$\gamma = \arccos\left(\frac{(\hat{\boldsymbol{x}}_1^{(\emptyset)})^\top \hat{\boldsymbol{x}}_1^{(c)}}{\|\hat{\boldsymbol{x}}_1^{(\emptyset)}\|_2 \|\hat{\boldsymbol{x}}_1^{(c)}\|_2}\right), \tag{57}$$

$$\gamma_\omega = \text{threshold}((\omega - 1)\gamma, \pi/3), \tag{58}$$

$$\hat{\boldsymbol{x}}_{1,\omega} = \cos(\gamma_\omega)\hat{\boldsymbol{x}}_1^{(c)} + \frac{\sin(\gamma_\omega)}{\sin(\gamma)}(\hat{\boldsymbol{x}}_1^{(c)} - \text{proj}_{\hat{\boldsymbol{x}}_1^{(\emptyset)}}(\hat{\boldsymbol{x}}_1^{(c)})), \tag{59}$$

$$\mathbf{v}_t = \frac{\hat{\boldsymbol{x}}_{1,\omega} - (1 - \sigma_{\min})\boldsymbol{x}_t}{1 - (1 - \sigma_{\min})t}, \tag{60}$$

$$\boldsymbol{x}_{t+\Delta t} = \mathbf{v}_t\Delta t + \boldsymbol{x}_t. \tag{61}$$

## G. Futher Experiment

### G.1. Fine Grained Experimental Results

We present the fine-grained experimental results of CFG and CFG++ here. Around the maximum value of the coarse-grained results, the step size is reduced to identify better-performing $\omega$ values for the reference algorithms. Notably, ADG **without** fine-grained hyperparameter tuning consistently outperforms the reference algorithms across the metrics and maintains its superiority over a wide range of $\omega$. This is attributed to its emphasis on angular domain guidance.

*Table 5.* Results of 10 NFE generation with SD v3.5 (d=38) on COCO10k (CFG++, $\lambda = \omega/12.5$).

| VALUE | $\omega = 1$ | $\omega = 1.5$ | $\omega = 2$ | $\omega = 2.5$ | $\omega = 3$ |
|---|---|---|---|---|---|
| CLIP↑ | 0.296 | 0.311 | **0.315** | **0.315** | 0.314 |
| IR↑ | -0.161 | 0.456 | **0.574** | 0.477 | 0.300 |
| FID↓ | 18.1 | 17.3 | **17.1** | 17.2 | 17.2 |

| VALUE | $\omega = 4$ | $\omega = 6$ | $\omega = 8$ | $\omega = 10$ | BEST |
|---|---|---|---|---|---|
| CLIP↑ | 0.306 | 0.282 | 0.268 | 0.257 | 0.315 |
| IR↑ | -0.148 | -0.980 | -1.290 | -1.509 | 0.574 |
| FID↓ | 18.0 | 19.8 | 20.6 | 21.4 | 17.1 |

## G.2. Evaluation of Generated Image Quality with LLM

In this section, we leverages GPT-4o to assess the quality of generated images from multiple perspectives. Our evaluation simultaneously considers three crucial aspects: image quality and authenticity, text-image alignment, and semantic consistency. As demonstrated in Table 6, the evaluation results reveal that ADG exhibits overwhelming advantages over CFG, particularly under large guidance weights. The superior performance of ADG can be attributed to its ability to maintain high-fidelity image generation while ensuring precise alignment with textual descriptions, even when subjected to strong guidance conditions. This multi-faceted evaluation approach provides a more robust and reliable assessment of the generated images.

*Table 6.* Evaluation of Generated Image Quality with GPT-4o.

| $\omega$ | 2 | 4 | 6 | 8 | 10 |
|---|---|---|---|---|---|
| ADG better | **228(83.21%)** | **1284(85.31%)** | **4744(95.44%)** | **6276(97.96%)** | **8001(99.00%)** |
| CFG better | 46(16.79%) | 221(14.69%) | 179(4.56%) | 131(2.04%) | 81(1.00%) |
| Similar | 9726 | 8495 | 5077 | 3593 | 1918 |

## G.3. Evaluation with Complex Text Prompts

To evaluate ADG under more complex prompts, we curated 500 complex text prompts using GPT. Our experiments show that ADG significantly outperforms CFG under these conditions. For guidance weight $\omega = 8$, ADG achieves a CLIP score of 0.355 and an IR score of 1.566 while CFG scores lower with a CLIP score of 0.338 and an IR score of 0.766. Examples of the generated outputs are partially illustrated in Figure 7.

## G.4. Qualitative Results at Extreme Guidance Weight

To further evaluate the robustness of ADG under extreme guidance conditions, we conduct additional experiments using a guidance weight of $\omega = 20$. Figure 8 presents representative samples generated by both CFG and ADG.

As shown, CFG exhibits severe artifacts, including oversaturation, unnatural textures, and significant semantic drift from the input prompts. In contrast, ADG consistently produces visually coherent and semantically faithful images, even under such high guidance weight. These results further validate the stability and effectiveness of ADG in extreme settings.

## G.5. Results with Different Guidance Weight and NFE

We show the methods with different guidance weights and NFEs as in Figures 9 to 11. Notably, the ADG and CFG algorithms maintain relative stability across various NFEs, while the CFG++ algorithm does not. For instance, with a guidance weight of 0.2, the CFG++ algorithm exhibits a relatively normal saturation at NFE=10. However, at NFE=40, it shows noticeable oversaturation, and similar trends can be observed with guidance weights of 0.3, 0.4, and 0.5.

This behavior may be due to the CFG++ algorithm being equivalent to a time-varying guidance weight CFG algorithm. The time-varying guidance weight, influenced by the discrete step size, likely causes its instability relative to both the CFG and ADG algorithms.

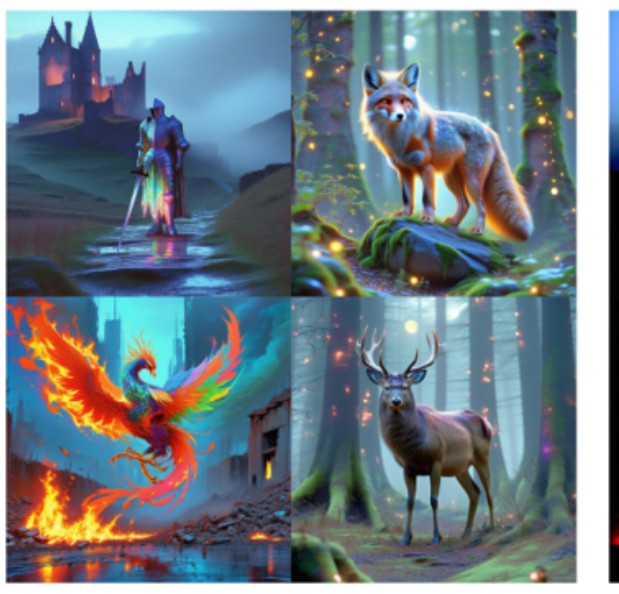 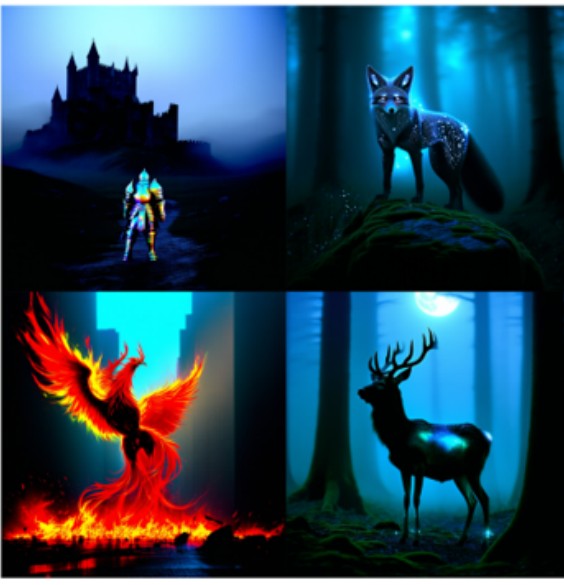

7.a ADG          7.b CFG

*Figure 7.* Comparison of ADG and CFG with complex text prompts under $\omega = 8$. While CFG shows notable degradation, ADG with normalization maintains stable performance.

Another noteworthy point is that, for different NFE values, ADG consistently generates images that align well with the textual prompt while maintaining relative stability in image content under the same random seed. In contrast, CFG and CFG++ exhibit significant variations in image content across different NFE values. This further highlights the relative stability of our proposed algorithm.

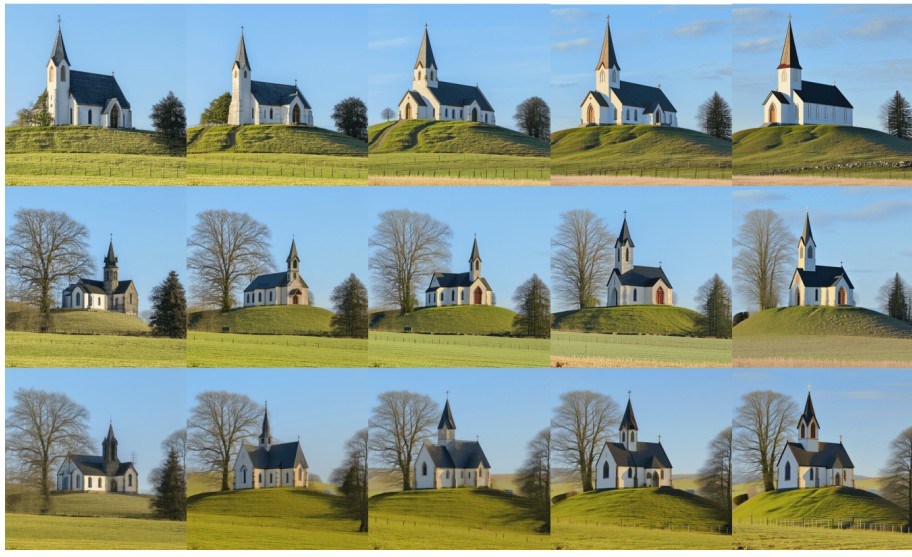

*Figure 9.* Generated images using ADG for the prompt "***The church is up on the hill in the country***" Rows correspond to different numbers of function evaluations (NFE): 40 (top), 20 (middle), and 10 (bottom). Columns represent increasing guidance weights: 2, 4, 6, 8, and 10 (from left to right). The same random seed is used across all generations.

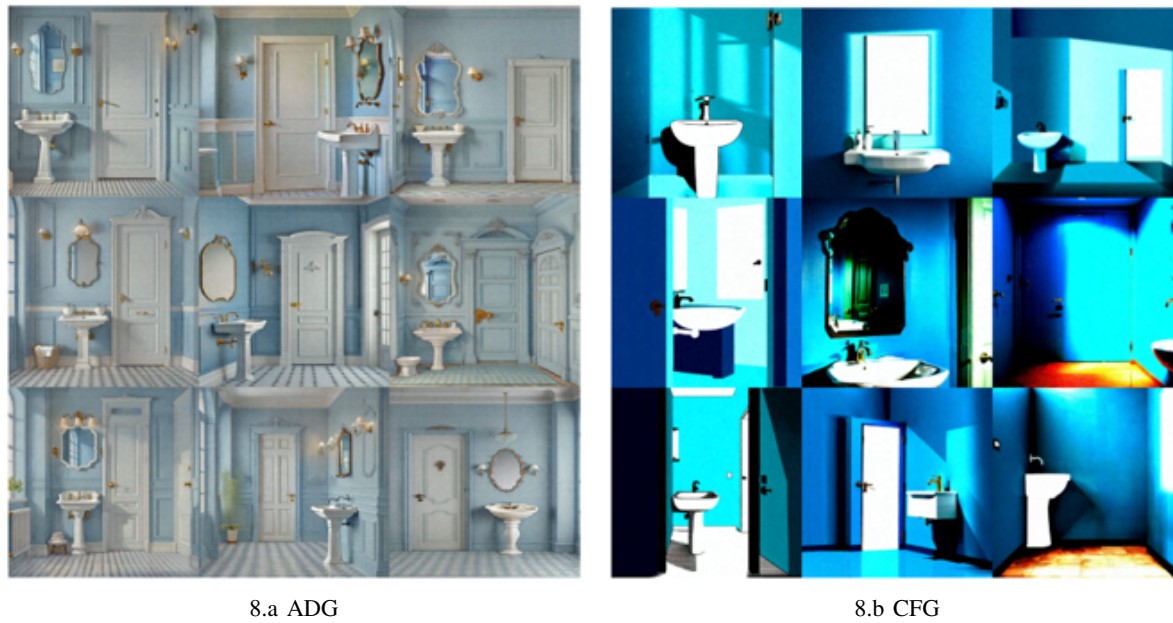



8.a ADG             8.b CFG



*Figure 8.* Comparison of ADG and CFG with complex prompts at $\omega = 20$. ADG maintains stability, while CFG produces distorted and misaligned results. The prompt is "A room with blue walls and a white sink and door".

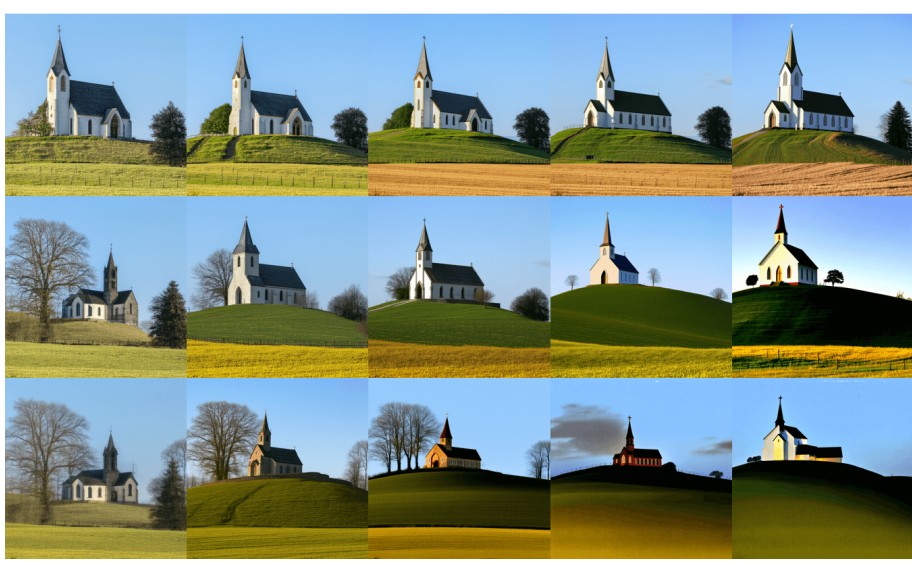

*Figure 10.* Generated images using CFG for the prompt "***The church is up on the hill in the country***" Rows correspond to different numbers of function evaluations (NFE): 40 (top), 20 (middle), and 10 (bottom). Columns represent increasing guidance weights: 2, 4, 6, 8, and 10 (from left to right). The same random seed is used across all generations.

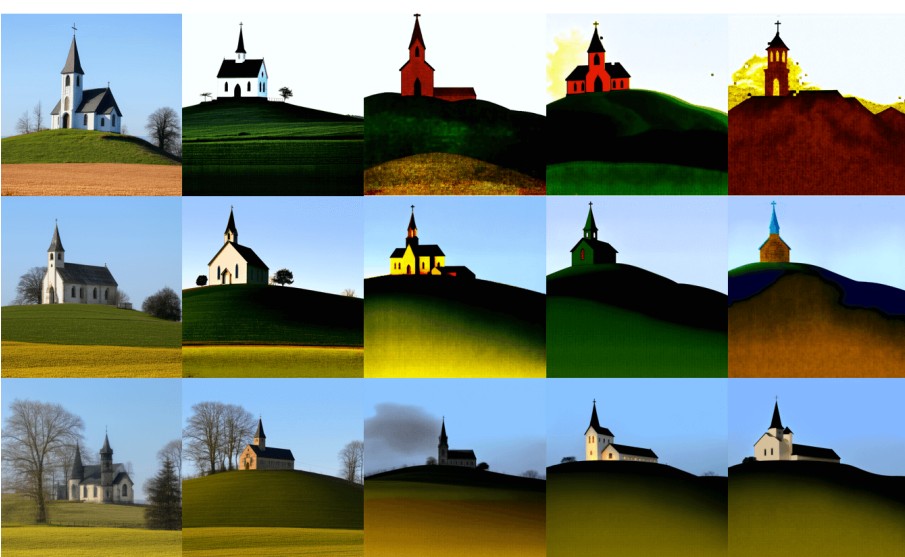

*Figure 11.* Generated images using CFG++ for the prompt "***The church is up on the hill in the country***" Rows correspond to different numbers of function evaluations (NFE): 40 (top), 20 (middle), and 10 (bottom). Columns represent increasing guidance weights: 0.1, 0.2, 0.3, 0.4, and 0.5 (from left to right). The same random seed is used across all generations.

### G.6. More Results on COCO datasets

In this section, we provide more images generated by ADG and reference algorithms for comparison as in Figures 12 to 14.

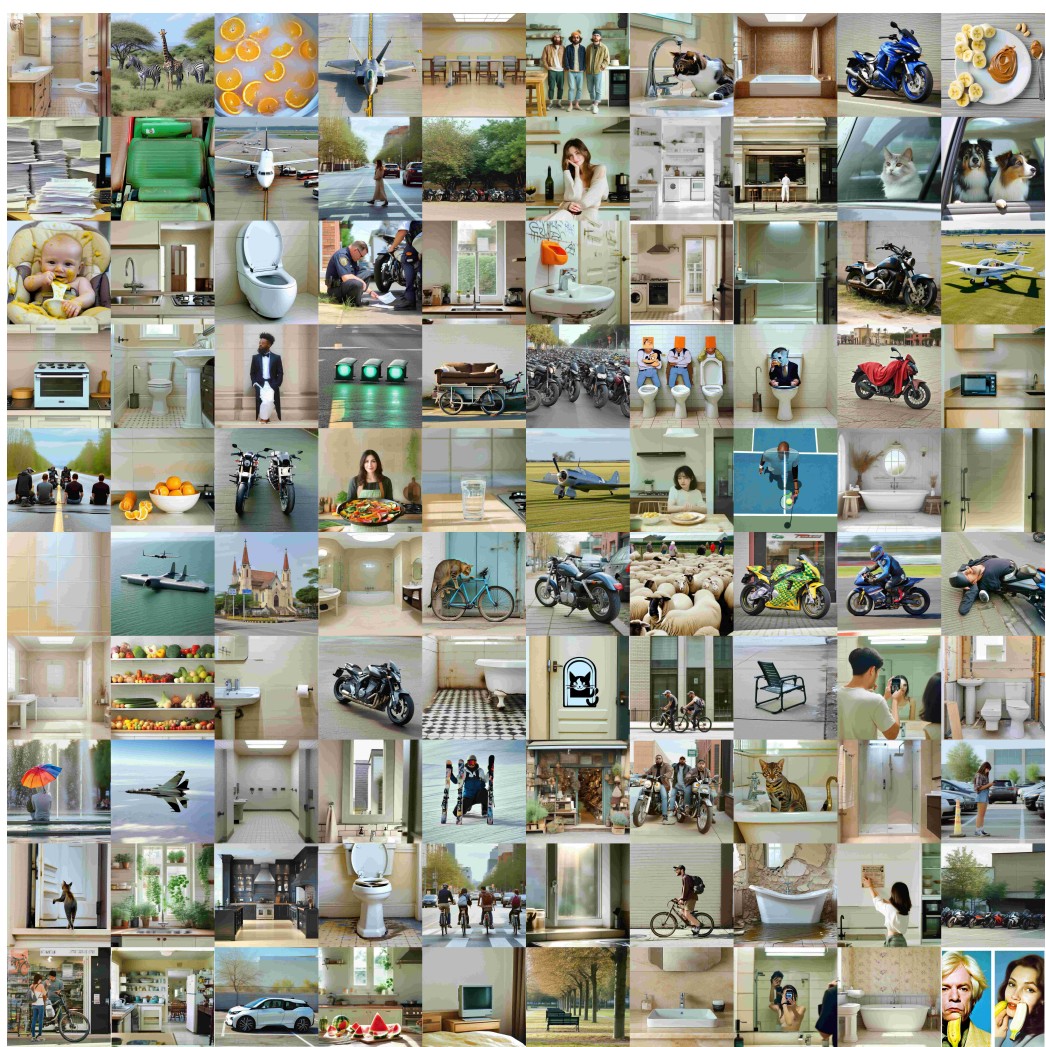

*Figure 12.* conditional COCO samples using ADG ($\omega = 6$, NFE=10).

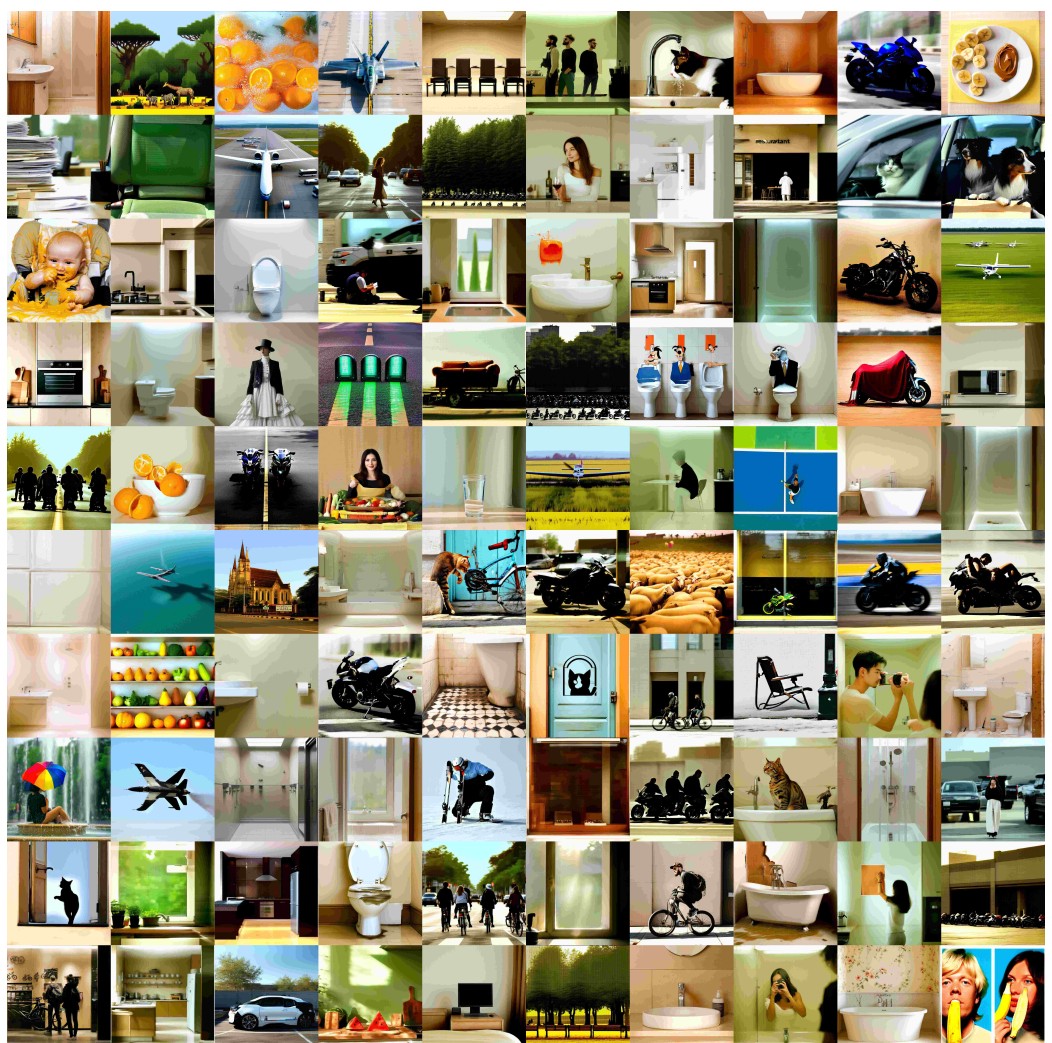

*Figure 13.* conditional COCO samples using CFG ($\omega = 6$, NFE=10).

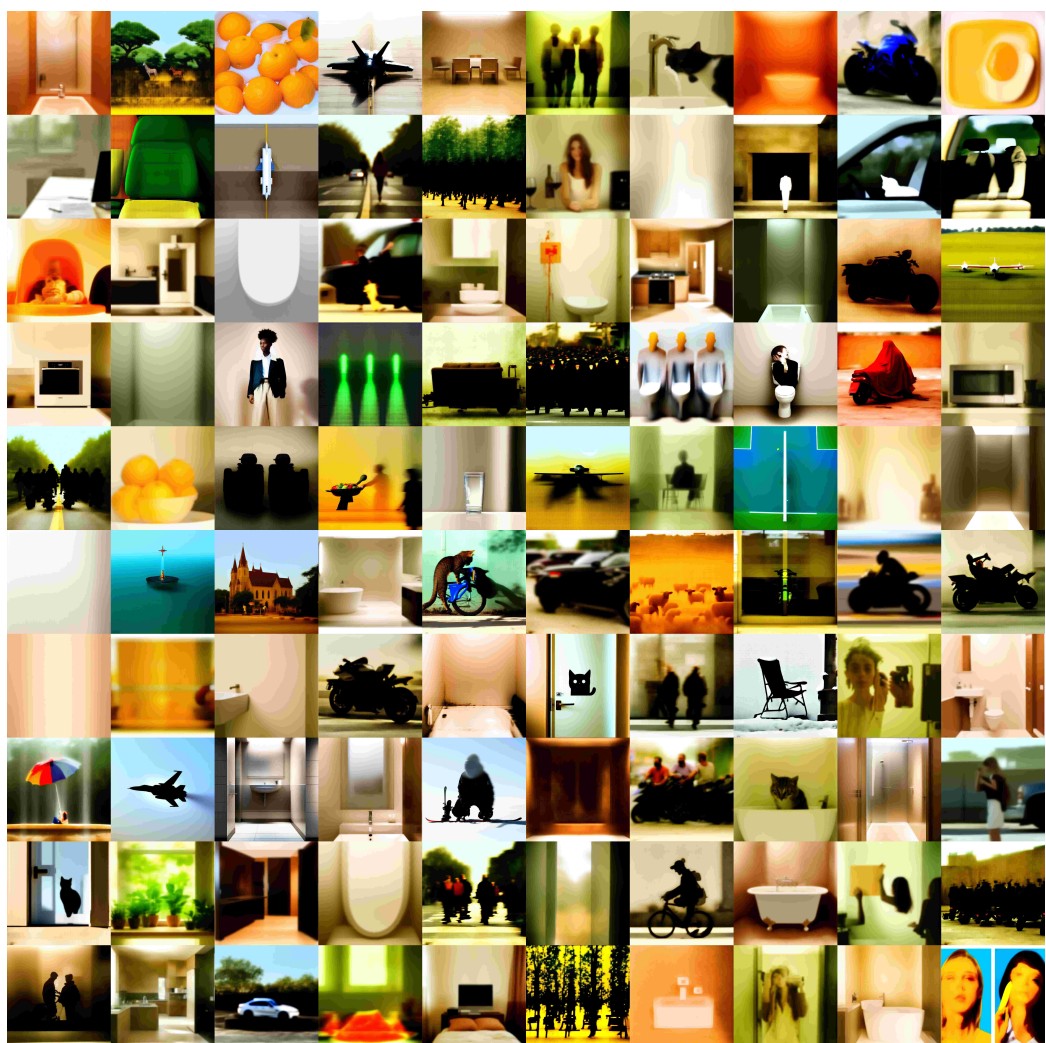

*Figure 14.* conditional COCO samples using CFG++ ($\lambda = 0.3$, NFE=10).

