# OpenReview forum: "Angle Domain Guidance: Latent Diffusion Requires Rotation Rather Than Extrapolation"
_ICML.cc/2025/Conference — ICML 2025 poster_

### Official Review · Reviewer_QHkd · 2025-03-13

**Overall Recommendation:** 3

**Summary:**

This paper introduces Angle Domain Guidance (ADG), a simple and effective sampling algorithm designed to improve the performance of text-to-image latent diffusion models, particularly under high guidance weights. The authors focus on the shortcomings of Classifier-Free Guidance (CFG), specifically its tendency to cause norm amplification in the latent space, which leads to color distortions and oversaturation in generated images. The paper provides a comprehensive theoretical analysis showing that CFG’s linear extrapolation mechanism results in sample norm inflation and anomalous diffusion phenomena. Based on this insight, ADG is proposed as an alternative that focuses on angular alignment rather than magnitude extrapolation in the latent space, ensuring better text-image alignment without sacrificing image fidelity. Experimental results on the COCO dataset demonstrate that ADG achieves superior performance in terms of CLIP Score, ImageReward, and FID across a wide range of guidance weights, outperforming both CFG and CFG++.

**Claims And Evidence:**

The paper claims that (1) CFG leads to significant color distortions at high guidance weights due to norm amplification in latent space, (2) ADG effectively mitigates these issues by controlling magnitude variation while enhancing angular alignment, and (3) ADG offers better text-image alignment, improved color fidelity, and superior perceptual quality. The evidence is compelling: both theoretical analyses and extensive empirical experiments support these claims. Figures and quantitative metrics (CLIP, ImageReward, FID) consistently demonstrate ADG’s robustness and effectiveness at high guidance weights, where CFG and CFG++ degrade. The ablation studies further substantiate the role of angular constraints in preventing catastrophic failures, solidifying the evidence behind the paper’s key claims.

**Essential References Not Discussed:**

N/A

**Experimental Designs Or Analyses:**

The experimental design is sound and systematic.
The authors test ADG across varying guidance weights ($\omega=2$  to $\omega=10$), compare against CFG and CFG++, and include ablation studies to validate the contribution of angular constraints and normalization. Experiments on both Stable Diffusion v3.5 (COCO dataset) and Stable Diffusion v2.1 with DPM-Solver highlight ADG’s generality. Key metrics are reported clearly, with ADG consistently outperforming baselines in ImageReward (often used as a proxy for human preference) and CLIP Score. Ablation studies demonstrate the necessity of angular constraints to avoid instability.

**Methods And Evaluation Criteria:**

The proposed ADG method is well-motivated, deriving from a sound analysis of the pitfalls of existing CFG. The authors introduce a geometrically inspired approach by emphasizing angular guidance in the latent space, consistent with the assumption that latent representations follow high-dimensional spherical Gaussian distributions. Evaluation criteria include standard and accepted metrics in text-to-image generation: CLIP Score (semantic alignment), ImageReward (human preference alignment), and FID (distributional similarity).
The authors run experiments on the COCO dataset using Stable Diffusion v3.5 and validate compatibility with advanced samplers such as DPM-Solver.
The evaluation is adequate, though it relies primarily on automated metrics without additional human evaluations, which are often necessary for perceptual alignment validation in generative modeling.

**Other Comments Or Suggestions:**

N/A

**Other Strengths And Weaknesses:**

The paper’s primary strength lies in its novel conceptual shift from linear extrapolation to angular guidance in latent diffusion sampling. ADG is a theoretically motivated and practically effective approach that mitigates common issues with CFG, especially at high guidance weights. The theoretical analysis is comprehensive and extends prior literature in meaningful ways. Additionally, the method is shown to be compatible with various samplers and diffusion frameworks, highlighting its flexibility.

On the weakness side, the empirical validation is narrowly focused on latent diffusion models (i.e., Stable Diffusion). While the authors acknowledge ADG’s heuristic nature and its dependence on the latent space structure of variational autoencoders, a deeper analysis of potential limitations when applied to non-latent diffusion models (e.g., pixel-space diffusion) would be helpful. Moreover, computational costs associated with ADG, particularly in high-dimensional spaces, are not discussed in detail.

**Questions For Authors:**

1. Does the similar trends of CFG also resides in pixel diffusion models?
2. The paper assumes that the latent space follows spherical Gaussian distribution, could the author elaborate or provide additional visualization to validate this claim?
3. While original CFG use linear interpolation, one natural way is to use spherical linear interpoliation (slerp) to perform angular guidance. Is there any reason why did not tried slerp as the design choice for angular guidance?

**Relation To Broader Scientific Literature:**

The proposed paper provides a method to improve image generation, while the method could be further investigated to diffusion models on other data, e.g., protein, text, etc., as the conditional generation with diffusion model plays a crucial role in various machine learning tasks.

**Theoretical Claims:**

The paper provides a theoretical framework analyzing the shortcomings of CFG. Specifically, Theorem 3.2 introduces the norm amplification effect of CFG, and Theorem 3.3 shows anomalous diffusion, which offers insight into how CFG might induce undesirable latent space behaviors at high guidance scale.
The authors extend beyond prior analyses by introducing the concept of surface classes and offering proofs in high-dimensional settings with multiple components.
However, while the theoretical arguments are sound, the practical extension of these results to complex real-world latent spaces (beyond Gaussian mixtures) could be discussed in more depth.

---

> ### Author Rebuttal · Authors · 2025-03-31
>
> Thank you for your positive comments. We provide our responses below.
> ### 1. **Clarification on Computational Costs of ADG**
> **Reviewer Concern:**
> The reviewer mentions that computational costs associated with ADG, particularly in high-dimensional spaces, are not discussed in detail.
>
>
> **Response:**
> We appreciate the reviewer’s observation. ADG introduces negligible computational overhead compared to standard sampling procedures. It does **not require additional neural network evaluations**; instead, its overhead consists solely of basic vector operations (e.g., normalization and angle clipping).
>
> To quantify this:
> - On **SD3.5-large** (latent dimensionality ≈ $2 \times 10^5$), the additional operations per inner loop cost around $1 \times 10^6$ FLOPs.
> - In contrast, one full sampling step requires approximately $9 \times 10^{13}$ FLOPs (measured using the `thop` package).
> - Empirical measurements over 100 generations on an A100 GPU show:
>   - **CFG**: Avg. generation time = **6.74s**
>   - **ADG**: Avg. generation time = **6.72s**
>
> This confirms that **ADG does not increase runtime**, and minor variation may be attributed to system noise.
>
> ### 2. Behavior of CFG in Pixel-Space Diffusion Models
> **Reviewer Concern:**
> The reviewer asks whether similar trends of norm amplification and image degradation occur in **pixel-space diffusion models**.
>
>
> **Response:**
> Yes, similar degradation patterns are observed in pixel-space diffusion models under high guidance weights. As in latent models, CFG-induced norm amplification pushes pixel values toward extreme ranges, leading to oversaturation and unnatural contrast. [visual example](https://files.catbox.moe/dqymkj.png)
>
> These observations reinforce our hypothesis that norm amplification is a core issue, not limited to latent spaces.
>
> ### 3. Justification for Assuming Spherical Gaussian Latent Space
> **Reviewer Concern:**
> The reviewer requests further elaboration or visualization to validate the assumption that the latent space follows a spherical Gaussian distribution.
>
>
> **Response:**
> During the pretraining of VAEs used in latent diffusion models, the latent space is regularized to approximate a standard multivariate Gaussian distribution through a Kullback-Leibler (KL) divergence loss term. We acknowledge that Gaussian priors in VAEs are often idealized, and real-world latent spaces may deviate from this assumption due to model imperfections or data complexity. However, this Gaussian prior remains a practical and effective approximation that helps explain why directional information carries more semantic meaning than magnitude information.
>
> Samples drawn from a high-dimensional Gaussian distribution tend to concentrate around a thin spherical shell, where the angular relationship between latent variables retains critical semantic information. Consequently, emphasizing angular alignment, as done in Angle-Domain Guidance (ADG), effectively preserves text-image consistency while mitigating the undesirable effects of norm amplification. This is also one of the reasons why we emphasize that, although ADG is theoretically inspired and has some theoretical guarantees, it remains a heuristic algorithm.
> ### 4. Choice of ADG over Spherical Linear Interpolation (SLERP)
> **Reviewer Concern:**
> The reviewer suggests that using **spherical linear interpolation (SLERP)** could be a natural alternative to ADG for performing angular guidance and inquires why it was not considered.
>
>
> **Response:**
> While **SLERP** is an elegant mathematical alternative, it is not suitable for high guidance weights in our setting. The reasons are as follows:
> - SLERP inherently performs interpolation, not extrapolation. In cases where the guidance weight $\omega > 1$, SLERP **extrapolates** beyond the two endpoints, leading to uncontrolled norm growth.
> - Our proposed ADG method, by contrast, **constrains the norm** of the generated sample under high guidance weights, as demonstrated by **Proposition 4.1**.
> - Consider a simple example where:
> $$
> \hat x_{0, c} = [1, 0]^\top, \quad \hat x_{0, \emptyset} = [0.5, 0.01]^\top.
> $$
> Using SLERP with $\omega = 5$, the result is:
> $$
> \text{slerp}(\hat x_{0, \emptyset}, \hat x_{0, c}, 5) \approx [15.9, -1.2]^\top,
> $$
> which exhibits extreme sensitivity when the vectors are nearly aligned.
> In contrast, ADG mitigates this instability by ensuring angular alignment while maintaining controlled magnitude growth.
> We chose ADG over SLERP due to its norm control under high guidance weights, which is crucial for maintaining image fidelity.
>
>
> Once again, thank you for your constructive feedback and for considering our paper for acceptance. We will revise our paper accordingly.

---

### Official Review · Reviewer_FRgS · 2025-03-14

**Overall Recommendation:** 4

**Summary:**

This paper focuses on the problem of color distortions in the generated images when classifier-free guidance is set to a high value. This paper identifies that these distortions come from the amplification of sample norms in the latent space. To address this problem, this paper proposes Angle Domain Guidance (ADG) algorithm. ADG constrains magnitude variations while optimizing angular alignment, thereby mitigating color distortions while preserving the enhanced text-image alignment achieved at higher guidance weights. Experimental results demonstrate the effectiveness of ADG.

**Claims And Evidence:**

The main claim of this paper is to address the color distortion problem when classifier-free guidance is set to a high value. The experiment results successfully validate the claim.

**Essential References Not Discussed:**

The references are complete and comprehensive.

**Experimental Designs Or Analyses:**

See "Methods And Evaluation Criteria".

**Methods And Evaluation Criteria:**

The method and evaluation are pretty complete, but I still have some questions:
- I am curious about the performance of a higher CFG, e.g.,  larger than 20. This would further show the effectiveness of ADG.
- Since sdv3.5 is pretrained on text-to-image generation, it's better to show more results containing complex text prompts as guidance (instead of simple prompts that describe a single object). This would further show how text alignment becomes when we increase the guidance scale.
- More experiments on state-of-the-art models (e.g., Flux) would make the claim more solid.

**Other Comments Or Suggestions:**

See above.

**Other Strengths And Weaknesses:**

This paper addresses an important issue in this field and can lead to a broader impact on more general generation methods. The paper writing of this paper is clear and easy to follow.

**Questions For Authors:**

See above.

**Relation To Broader Scientific Literature:**

The contribution of this paper can be further applied to broader generation models such as motion generation and video generation, which would have a broader impact on the research field.

**Theoretical Claims:**

The theoretical claims are correct.

---

> ### Author Rebuttal · Authors · 2025-03-31
>
> Thank you for your positive comments. We provide our responses below.
>
> ### **1. Performance of Higher CFG Values**
>
> **Reviewer Comment:**
> > I am curious about the performance of a higher CFG, e.g., larger than 20. This would further show the effectiveness of ADG.
>
> **Response:**
> We appreciate this insightful suggestion. We have conducted additional experiments with higher guidance weights, specifically at weight = 20. The results demonstrate that ADG maintains stable performance even under extreme guidance conditions, whereas CFG suffers from severe color distortions and semantic misalignment.These results highlight the robustness of ADG at high guidance levels. We will include qualitative visualizations at weight = 20 in the supplementary material.
> [View Images](https://files.catbox.moe/lnh69r.png)
>
>
> ### **2. Evaluation with Complex Text Prompts**
>
> **Reviewer Comment:**
> > Since sdv3.5 is pretrained on text-to-image generation, it's better to show more results containing complex text prompts as guidance (instead of simple prompts that describe a single object). This would further show how text alignment becomes when we increase the guidance scale.
>
> **Response:**
> Thank you for this helpful suggestion. To evaluate ADG under more complex prompts, we curated 500 complex text prompts using GPT.
> Our experiments show that ADG significantly outperforms CFG under these conditions. For guidance weight = 8:
> - ADG achieves a CLIP score of 0.355 and an IR score of 1.566
> - CFG scores lower with a CLIP score of 0.338 and an IR score of 0.766
>
> These results confirm that ADG preserves semantic alignment and image quality even with complex instructions. We will include detailed metrics and visual comparisons in the supplementary material. [View Images](https://files.catbox.moe/p5ayb9.png)
>
>
> ### **3. Experiments on State-of-the-Art Models (e.g., Flux)**
>
> **Reviewer Comment:**
> > More experiments on state-of-the-art models (e.g., Flux) would make the claim more solid.
>
> **Response:**
> Thank you for raising this important point. We initially considered incorporating results from state-of-the-art open-source models, including **Stable Diffusion 3.5 large** and **Flux.1 [dev]**. However, during our experiments, we discovered that Flux.1 [dev] applies guidance weight as a **model input** rather than through the conventional CFG mechanism.
>
> Upon further investigation of the Flux architecture and related blog posts, we learned that Flux.1 [dev] is derived via guidance distillation from Flux.1 [pro], which is unfortunately **not open-source**. Since access to Flux.1 [pro] is required to modify the guidance mechanism, it was impossible for us to conduct experiments on the Flux family.
>
> Once again, thank you for your constructive feedback and for considering our paper for acceptance. We will revise our paper accordingly.

---

> > ### Comment · Reviewer_FRgS · 2025-04-04
> >
> > The rebuttal effectively addressed and resolved the concerns I had. After reviewing the response in detail, I feel that my initial reservations have been adequately clarified. As a result, I am satisfied with the explanation provided and will maintain my original rating of "accept."

---

> > > ### Author Response · Authors · 2025-04-07
> > >
> > > Thank you very much for your kind feedback and for taking the time to carefully review our rebuttal. We truly appreciate your thoughtful evaluation and are glad to know that our clarifications addressed your concerns.

---

### Official Review · Reviewer_pHtg · 2025-03-17

**Overall Recommendation:** 4

**Summary:**

The paper presents angle domain guidance (ADG), an alternative to classifier-free guidance (CFG) for conditional diffusion models. The key observation is that CFG leads to excessively large sample norms, causing oversaturated colors in the generated images. The paper claims that this is a result of CFG's linear extrapolation scheme in the latent space, presents an analysis on norm magnitude in a simplified setting where the target distribution is a Gaussian mixture model, and proposes to instead extrapolate in the angular domain to align the directions of latents, thereby effectively controlling norm magnitude. The experimental results demonstrate that ADG outperforms CFG given large guidance weights.

**Claims And Evidence:**

- The paper claims that CFG amplifies sample norm, which is indicative of poor sample quality. This claim is substantiated by empirical evidence (Figure 2) showing that norm magnitude is proportional to guidance weight, and color values are positively correlated with norm magnitude. The paper further presents a theoretical analysis (Section 3) which echoes the empirical findings. The analysis is performed in a simplified setting, where the target distribution is assumed to be a mixture of Gaussians. Nevertheless, this analysis sheds light on the challenge and informs the design of the new guidance algorithm.

- The paper claims that ADG mitigates norm amplification, thereby enhancing sample quality. This is backed by theoretical (Proposition 4.1) and experimental results (Section 5). To further strengthen this claim, I encourage the authors to plot norm against guidance weight (similar to Figure 2a) to find out whether ADG can empirically control sample norm as suggested by the theoretical result.

- The paper claims that ADG generalizes across samplers and can work with flow-based models. This is supported by experimental results with the DPM sampler (Table 3) and an extension of ADG that fits the flow-matching formulation (Appendix F). However, no experimental results are provided for flow-based models.

**Essential References Not Discussed:**

I am not familiar with recent literature on the analysis of diffusion model sampling. My impression is that the paper has adequately covered the most related works, given the quite extensive discussion in the supplementary material.

**Experimental Designs Or Analyses:**

- The experiments compare ADG against two baselines, namely CFG and CFG++, under varying guidance weights. An ablation study is performed to understand what design choices impact the performance of ADG. I do not have major concerns about the experimental setting, although I encourage the authors to showcase qualitative results on more diverse text prompts.

- One potential baseline could be CFG with normalization. That is, performing CFG at each sampling step, followed by normalizing and re-scaling the estimated x_0's, similar to Algorithm 6. This could be a simpler remedy to the norm amplification issue compared to ADG.

**Methods And Evaluation Criteria:**

- The proposed method aims to address a limitation of CFG, namely poor sample quality under large guidance weights. The method is motivated by a theoretical analysis and is both simple and effective.

- The benchmarks cover multiple aspects of sample quality, namely image quality (FID), image-text alignment (CLIP score), and human preference (IR). The same set of metrics has been used by the community for the evaluation of text-to-image diffusion models. Additional qualitative results further highlight the strength of the proposed method.

**Other Comments Or Suggestions:**

N/A

**Other Strengths And Weaknesses:**

- Overall, the paper is well motivated and clearly written.

- I encourage the authors to expand on what they mean by "non-commutativity of the tilting process with the
forward process" (L126-127) and briefly explain why Equation 10 holds.

**Questions For Authors:**

N/A

**Relation To Broader Scientific Literature:**

The proposed method fits in the literature of conditional diffusion models and provides an alternative to the widely used CFG for improving sample quality and condition following.

**Theoretical Claims:**

The paper provides a theoretical analysis on the impact of CFG on sample norms. The analysis is performed in a simplified setting, where the target distribution is assumed to be a mixture of Gaussians. I did not carefully check the correctness of the proofs, yet the claims make intuitive sense and are partially justified by the experimental results.

---

> ### Author Rebuttal · Authors · 2025-03-31
>
> Thank you for your positive comments. We provide our responses below.
>
> ### **1. Plot norm against guidance weight for ADG**
>
> **Reviewer Comment:**
> > To further strengthen this claim, I encourage the authors to plot norm against guidance weight (similar to Figure 2a) to find out whether ADG can empirically control sample norm as suggested by the theoretical result.
>
> **Response:**
> We appreciate this suggestion. We have added a new plot to **Figure 2a**, showing the norm against guidance weight for ADG.
> [View Figure 2a](https://files.catbox.moe/mtxaye.png)
>
> The experimental results indicate that **ADG effectively controls sample norms**, preventing norm amplification even at high guidance weights.
>
>
> ### **2. Experimental results for flow-based models**
>
> **Reviewer Comment:**
> > However, no experimental results are provided for flow-based models.
>
> **Response:**
> Thank you for highlighting this point. We would like to clarify that our primary results were obtained on **SDv3.5**, and Stable Diffusion series starting from version 3 employs flow-based models[a].
>
> To avoid any ambiguity, we will emphasize this point more explicitly in the camera-ready version.
>
>
> ### **3. CFG with normalization as a potential baseline**
>
> **Reviewer Comment:**
> > One potential baseline could be CFG with normalization. That is, performing CFG at each sampling step, followed by normalizing and re-scaling the estimated x_0's, similar to Algorithm 6. This could be a simpler remedy to the norm amplification issue compared to ADG.
>
> **Response:**
> This is a good suggestion. Since our motivation stems from the observation that **angular information in the x_0 domain better aligns with semantics**, while **norm amplification** under high CFG weights degrades sample quality, we designed ADG to enhance angular consistency. The method proposed by the reviewer is a simple way to control sample norms.
>
> We implemented and tested this potential baseline (CFG and normalization in the x_0 domain). Experimental results show that although this method performs slightly worse than ADG, it significantly outperforms the original CFG. Our analysis suggests that the angle between the predicted x_0 and the unconditional prediction under this baseline is smaller than the angle between the conditional prediction and the unconditional prediction in ADG, limiting semantic alignment despite effective norm control.
> [View comparison](https://files.catbox.moe/e0scpg.png)
>
> ### **4. Explanation of non-commutativity and Equation 10**
>
> **Reviewer Comment:**
> > I encourage the authors to expand on what they mean by "non-commutativity of the tilting process with the forward process" (L126-127) and briefly explain why Equation 10 holds.
>
> **Response:**
> Due to space constraints, we could not elaborate on this point in the main text. However, as noted in [c], **non-commutativity** means that the weighted gradient used in CFG does not align with the true gradient of the tilted target distribution for t>0.
>
> To illustrate this, consider a simple example where:
> - The unconditional distribution is modeled as a Gaussian with zero mean and variance 10.
> - The conditional distribution is another Gaussian with zero mean and unit variance.
>
> When applying classifier-free guidance (CFG) with a guidance weight of 2, the tilted distribution becomes a Gaussian with:
> - Zero mean.
> - A variance of 10/19.
>
> After applying the forward process, which introduces Gaussian noise over time (considering a Variance Exploding SDE, i.e., VE-SDE), the resulting distribution is convolved with a Gaussian kernel whose variance increases with time.
> To simplify the computation, assume that the time-dependent Gaussian kernel at a particular time t* is modeled as a Gaussian with variance 1.
>
> In this setting:
> - The gradient of the CFG-modified distribution is a weighted sum of the gradients of the conditional and unconditional distributions, which results in a gradient proportional to **-10/11 \* x**.
> - However, this gradient does **not** match the gradient of the tilted distribution obtained after applying the forward process, which is proportional to **-19/29 \* x**, highlighting the **non-commutativity** between the tilting process and the forward process.
>
> Once again, thank you for your constructive feedback and for considering our paper for acceptance. We will revise our paper accordingly.
>
> ## **References**
>
> - [a] [Stable Diffusion 3](https://stability.ai/news/stable-diffusion-3)
> - [b] Exploring diffusion and flow matching under generator matching. arXiv
> - [c] What does guidance do? A fine-grained analysis in a simple setting. NeurIPS 2024.

---

### Official Review · Reviewer_6TQ2 · 2025-03-17

**Overall Recommendation:** 3

**Summary:**

This paper attempts to analyze the distributions of conditional generation vs. unconditional generation, and claims that in some occasions the direction of classifier-free guidance may be "abnormal", i.e., leads to low-probability density areas, and proposes "Angle-Domain Guidance Sampling" (ADG) as a remedy.

UPDATE after author response:
The author response makes me understand the paper and some statements better. In particular, I appreciate the authors provide extra comparative examples with APG. I'd like to raise the rating to weak accept.

**Claims And Evidence:**

1. After two hours of reading, I feel very confused about the theoretical derivations, especially I don't understand why they lead to the "Angle-Domain Guidance Sampling" (ADG). In particular,
    1) The theorems and lemmas in section 3.2 don't seem to lead to the claim that "CFG leads to excessively large norms of features".
    2) More confusingly, when the authors present motivations of ADG, they said "The focus on magnitude differences is secondary, and in the case of high guidance weights, it can even be detrimental", so this means that large norms are not important, but angles are important? Then why in the abstract, "these distortions stem from the amplification of sample norms in the latent space"?

**Essential References Not Discussed:**

[a] Eliminating Oversaturation and Artifacts of High Guidance Scales in Diffusion Models. ICLR 2025.

**Experimental Designs Or Analyses:**

The empirical evaluation seems to be fine. However, perhapse the most important baseline is adaptive projected guidance (APG) [a], which is not mentioned or compared with.

[a] Eliminating Oversaturation and Artifacts of High Guidance Scales in Diffusion Models. ICLR 2025.

**Methods And Evaluation Criteria:**

1. As listed in "Claims And Evidence", the derivations are confusing and I don't understand why ADG is a logical consequence of the theorems.
2. The empirical evaluation seems to be fine.

**Other Comments Or Suggestions:**

1. Please center your derivations around your main claims/methods.
2. Preferablly, illustrate math derivations with actual examples of generation.

**Other Strengths And Weaknesses:**

N/A

**Questions For Authors:**

N/A

**Relation To Broader Scientific Literature:**

ADG seems to be highly similar to adaptive projected guidance (APG) [a], although ADG is obfuscated by some non-essential transformations in Algorithm 1 (arccos, discount then cos). Whereas APG is not cited (It's not a concurrent work since APG has been on arxiv since Oct 2024).

[a] Eliminating Oversaturation and Artifacts of High Guidance Scales in Diffusion Models. ICLR 2025.

**Theoretical Claims:**

Same as those listed in "Claims And Evidence".

---

> ### Author Rebuttal · Authors · 2025-03-31
>
> We sincerely thank the reviewer for their thoughtful comments. While the overall evaluation was critical, your constructive feedback is highly valuable and will help us improve both the clarity and impact of our work. Below, we provide detailed responses to your key concerns.
>
> ### 1. Comparison with Adaptive Projected Guidance
> **Reviewer Concern:**
> The reviewer notes the absence of discussion on Adaptive Projected Guidance (APG) [ICLR 2025], suggests high similarity with our proposed ADG method, and questions whether ADG offers substantive novelty.
> ﻿
>
> **Response:**
> We appreciate the reviewer for pointing out this important and timely work. APG appeared on arXiv in October 2024, shortly before our submission, and was therefore not included in our original analysis. We will include appropriate citations and a dedicated comparison section in the final version.
>
> To clarify the distinctions between ADG and APG, we summarize the differences below:
> ﻿
> 1. **Attribution of Image Degradation:**
> - APG attributes image oversaturation and degradation to the **parallel component** of the difference vector $\Delta \hat x_0$ relative to $\hat x_{0, c}$, denoted as $\Delta \hat x_0^\parallel$.
> - In contrast, we attribute oversaturation and distortion to the **large norm** of $\hat x_{0, CFG}$:
> $$
> \hat x_{0, CFG} = \hat x_{0, c} + (\omega - 1) \Delta \hat x_0.
> $$
> Removing the parallel component slows norm growth but **fails to address norm amplification**. Additional experiments show that normalizing $\hat x_{0, CFG}$ mitigates image degradation more effectively.
> [Visual example](https://files.catbox.moe/2msq2b.png)
> ﻿
> 2. **Algorithmic Difference:**
> - APG modifies $\Delta \hat x_0$ in CFG:
> $$
> \Delta \hat x_{0, APG} = \eta \Delta \hat x_0^\parallel + \Delta \hat x_0^\perp, \quad \hat x_{0, APG} = \hat x_{0, c} + \omega \Delta \hat x_{0, APG},
> $$
> where $\eta < 1$. However, APG retains **linear enhancement**, leading to norm growth at high guidance weights.
> - ADG, in contrast, performs angular-domain updates, effectively constraining norm growth (Proposition 4.1).
> [ADG vs. APG Visual](https://files.catbox.moe/1nnve4.png)
> ﻿
> 3. **Empirical Performance and Stability:**
>
> Under high guidance weights, **ADG successfully mitigates oversaturation and artifacts**, whereas **APG-generated images exhibit significant oversaturation**, which reduces text-image alignment. [visual examples](https://files.catbox.moe/c76emo.png)
>    ADG outperforms CFG and CFG++ in both alignment (CLIP) and human preference metrics (ImageReward), especially under high guidance weights.  [quantitative results](https://files.catbox.moe/cjzl1o.png)
>
> ﻿
> ### 2. Clarification on Theoretical Derivations and Algorithm Motivation
> **Reviewer Concern:**
> The reviewer seeks clarification on how the theoretical derivation leads to the conclusion that "CFG leads to excessively large norms of features." Additionally, they express confusion about the connection between the theoretical derivations and the proposed method. The reviewer also suggests illustrating the mathematical derivations with concrete generation examples where possible.
> ﻿
> **Response:**
> Theorem 3.2 shows that **linear extrapolation in CFG** shifts samples toward the outer regions of the unconditional distribution, increasing the norms of the latent variables (features).
> ﻿
> The paper's logical flow is:
> 1. **CFG Limitation:** Norm amplification and anomalous diffusion for surface-class samples (Theorem 3.2 and Theorem 3.3).
> 2. **Source of Norm Amplification:** Linear enhancement of $\hat{x}_0$ leads to larger norms.
> 3. **ADG Motivation:** Angular-domain updates mitigate norm amplification (Proposition 4.1).
>
> Fig. 3 illustrates this phenomenon with a Gaussian mixture model. The green, blue, and orange clusters represent surface classes, while the red cluster denotes non-surface classes. Higher guidance weights push surface-class samples away from the unconditional distribution, amplifying norms.
> ﻿
> ---
> ﻿
> ### 3. Clarification on “Magnitude Differences Are Secondary” Statement
> **Reviewer Concern:**
> The reviewer finds the statement *"The focus on magnitude differences is secondary, ..."* ambiguous.
> ﻿
>
> **Response:**
> We appreciate the reviewer's feedback. The intended insight is:
> ﻿
> - **Linear enhancement** methods (e.g., CFG) modify both the magnitude and direction of $\hat{x}_0$.
> - While **magnitude enhancement** may improve semantic alignment slightly, it becomes detrimental at high guidance weights due to excessive norm amplification, resulting in image degradation.
> #### Revised Statement:
> "Linear enhancement methods simultaneously modify the norm and direction of $\hat{x}_0$. While norm adjustments may marginally improve semantic alignment, excessive norm amplification at high guidance weights becomes detrimental, resulting in oversaturation and distortion."
>
> We appreciate your feedback and will revise the paper to clarify contributions and address concerns.

---

> > ### Comment · Reviewer_6TQ2 · 2025-04-06
> >
> > Now I understand the paper and some statements better. In particular, I appreciate the authors provide extra comparative examples with APG. I'd like to raise the rating to weak accept. In the meantime, I encourage the authors to release the source code of ADG for reviewers to verify. If my own tests agree with the claims in the paper, I'd like to further raise the rating to accept.

---

> > > ### Author Response · Authors · 2025-04-07
> > >
> > > Thank you very much for your thoughtful and constructive feedback, and for your willingness to raise the rating to “weak accept.” We greatly appreciate your recognition of our additional comparisons, including the evaluation with APG.
> > >
> > > Regarding your suggestion on code availability, we would like to clarify that, as mentioned in the **final paragraph of the Introduction in the manuscript**, the source code has been made publicly accessible via the anonymous repository: [https://anonymous.4open.science/r/ADGuidance/](https://anonymous.4open.science/r/ADGuidance/). The core implementation of the ADG algorithm used in our experiments is located in the file `method/ADG_SD3.py`, under the function `ADG_SD3`.
> > >
> > > In addition, the repository provides:
> > > - a `README.md` file with instructions on how to use the ADG method, and
> > > - a `vis.ipynb` notebook that reproduces most of the visualizations shown in the main paper.
> > >
> > > Please note that, upon acceptance, the anonymous repository will be replaced with a permanent public GitHub repository to ensure long-term accessibility and reproducibility.
> > >
> > > We sincerely appreciate your attention to the reproducibility of research and your encouragement for open-sourcing. If your own tests confirm our reported results, we would be truly grateful for your further consideration in raising the rating to “accept.”

---

### Decision · Program_Chairs · 2025-05-01

**Decision:**

Accept (poster)

**Comment:**

This paper proposes Angle-Domain Guidance Sampling (ADG), a new guidance method focusing on angular alignment rather than magnitude extrapolation in the latent space.

Strengths

* Clear motivation from both theoretical and empirical analysis of existing method (i.e., CFG)

* Effective method derived from analysis

* Strong empirical evaluation

Weaknesses

* Limited to latent diffusion models

The authors have addressed the reviewers’ concerns in the rebuttal, and I believe this paper presents ADG as a compelling guidance method, backed by comprehensive experiments and clear presentation, making it an impactful contribution to the field.